# THE SURPRISING COMPUTATIONAL POWER OF NONDETERMINISTIC STACK RNNS

**Brian DuSell and David Chiang**
Department of Computer Science and Engineering
University of Notre Dame
{bdusell1,dchiang}@nd.edu

## ABSTRACT

Traditional recurrent neural networks (RNNs) have a fixed, finite number of memory cells. In theory (assuming bounded range and precision), this limits their formal language recognition power to regular languages, and in practice, RNNs have been shown to be unable to learn many context-free languages (CFLs). In order to expand the class of languages RNNs recognize, prior work has augmented RNNs with a nondeterministic stack data structure, putting them on par with pushdown automata and increasing their language recognition power to CFLs. Nondeterminism is needed for recognizing all CFLs (not just deterministic CFLs), but in this paper, we show that nondeterminism and the neural controller interact to produce two more unexpected abilities. First, the nondeterministic stack RNN can recognize not only CFLs, but also many non-context-free languages. Second, it can recognize languages with much larger alphabet sizes than one might expect given the size of its stack alphabet. Finally, to increase the information capacity in the stack and allow it to solve more complicated tasks with large alphabet sizes, we propose a new version of the nondeterministic stack that simulates stacks of vectors rather than discrete symbols. We demonstrate perplexity improvements with this new model on the Penn Treebank language modeling benchmark.

## 1 INTRODUCTION

Standard recurrent neural networks (RNNs), including simple RNNs (Elman, 1990), GRUs (Cho et al., 2014), and LSTMs (Hochreiter & Schmidhuber, 1997), rely on a fixed, finite number of neurons to remember information across timesteps. When implemented with finite precision, they are theoretically just very large finite automata, restricting the class of formal languages they recognize to regular languages (Kleene, 1951). In practice, too, LSTMs cannot learn simple non-regular languages such as $\{w\#w^R \mid w \in \{0,1\}^*\}$ (DuSell & Chiang, 2020). To increase the theoretical and practical computational power of RNNs, past work has proposed augmenting RNNs with stack data structures (Sun et al., 1995; Grefenstette et al., 2015; Joulin & Mikolov, 2015; DuSell & Chiang, 2020), inspired by the fact that adding a stack to a finite automaton makes it a pushdown automaton (PDA), raising its recognition power to context-free languages (CFLs).

Recently, we proposed the nondeterministic stack RNN (NS-RNN) (DuSell & Chiang, 2020) and renormalizing NS-RNN (RNS-RNN) (DuSell & Chiang, 2022), augmenting an LSTM with a differentiable data structure that simulates a real-time nondeterministic PDA. (The PDA is *real-time* in that it executes exactly one transition per input symbol scanned, and it is *nondeterministic* in that it executes all possible sequences of transitions.) This was in contrast to prior work on stack RNNs, which exclusively modeled deterministic stacks, theoretically limiting such models to deterministic CFLs (DCFLs), which are a proper subset of CFLs (Sipser, 2013). The RNS-RNN proved more effective than deterministic stack RNNs at learning both nondeterministic and deterministic CFLs.

In practical terms, giving RNNs the ability to recognize context-free patterns may be beneficial for modeling natural language, as syntax exhibits hierarchical structure; nondeterminism in particular is necessary for handling the very common phenomenon of syntactic ambiguity. However, the RNS-RNN's reliance on a PDA may still render it inadequate for practical use. For one, not all phenomena in human language are context-free, such as cross-serial dependencies. Secondly, the RNS-RNN's

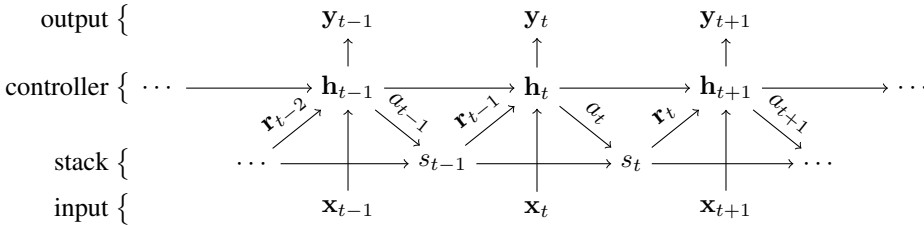

Figure 1: Conceptual diagram of the RNN controller-stack interface, unrolled across a portion of time. The LSTM memory cell $\mathbf{c}_t$ is not shown.

computational cost restricts it to small stack alphabet sizes, which is likely insufficient for storing detailed lexical information. In this paper, we show that the RNS-RNN is surprisingly good at overcoming both difficulties. Whereas an ordinary weighted PDA must use the same transition weights for all timesteps, the RNS-RNN can update them based on the status of ongoing nondeterministic branches of the PDA. This means it can coordinate multiple branches in a way a PDA cannot—for example, to simulate multiple stacks, or to encode information in the *distribution* over stacks.

Our contributions in this paper are as follows. We first prove that the RNS-RNN can recognize all CFLs and intersections of CFLs despite restrictions on its PDA transitions. We show empirically that the RNS-RNN can model some non-CFLs; in fact it is the only stack RNN able to learn $\{w\#w \mid w \in \{0,1\}^*\}$, whereas a deterministic multi-stack RNN cannot. We then show that, surprisingly, an RNS-RNN with only 3 stack symbol types can learn to simulate a stack of no fewer than 200 symbol types, by encoding them as points in a vector space related to the distribution over stacks. Finally, in order to combine the benefits of nondeterminism and vector representations, we propose a new model that simulates a PDA with a stack of vectors instead of discrete symbols. We show that the new vector RNS-RNN outperforms the original on the Dyck language and achieves better perplexity than other stack RNNs on the Penn Treebank. Our code is publicly available.[1]

## 2 STACK RNNS

In this paper, we examine two styles of stack-augmented RNN, using the same architectural framework as in our previous work (DuSell & Chiang, 2022) (Fig. 1). In both cases, the model consists of an LSTM, called the *controller*, connected to a differentiable stack. At each timestep, the stack receives *actions* from the controller (e.g. to push and pop elements). The stack simulates those actions and produces a *reading* vector, which represents the updated top element of the stack. The reading is fed as an extra input to the controller at the next timestep. The actions and reading consist of continuous and differentiable weights so the whole model can be trained end-to-end with backpropagation; their form and meaning vary depending on the particular style of stack.

We assume the input $w = w_1 \cdots w_n$ is encoded as a sequence of vectors $\mathbf{x}_1, \cdots, \mathbf{x}_n$. The LSTM's memory consists of a hidden state $\mathbf{h}_t$ and memory cell $\mathbf{c}_t$ (we set $\mathbf{h}_0 = \mathbf{c}_0 = \mathbf{0}$). The controller computes the next state $(\mathbf{h}_t, \mathbf{c}_t)$ given the previous state, input vector $\mathbf{x}_t$, and stack reading $\mathbf{r}_{t-1}$:

$$(\mathbf{h}_t, \mathbf{c}_t) = \text{LSTM}\left((\mathbf{h}_{t-1}, \mathbf{c}_{t-1}), \begin{bmatrix} \mathbf{x}_t \\ \mathbf{r}_{t-1} \end{bmatrix}\right).$$

The hidden state generates the stack actions $a_t$ and logits $\mathbf{y}_t$ for predicting the next word $w_{t+1}$. The previous stack and new actions generate a new stack $s_t$, which produces a new reading $\mathbf{r}_t$:

$$a_t = \text{ACTIONS}(\mathbf{h}_t) \quad \mathbf{y}_t = \mathbf{W}_{\text{hy}}\mathbf{h}_t + \mathbf{b}_{\text{hy}} \quad s_t = \text{STACK}(s_{t-1}, a_t) \quad \mathbf{r}_t = \text{READING}(s_t).$$

Each style of stack differs only in the definitions of ACTIONS, STACK, and READING.

### 2.1 SUPERPOSITION STACK RNN

We start by describing a stack with deterministic actions—the superposition stack of Joulin & Mikolov (2015)—which we include because it was one of the best-performing stack RNNs we

---

[1] https://github.com/bdusell/nondeterministic-stack-rnn

investigated previously (2020; 2022). The superposition stack simulates a combination of partial stack actions by computing three new, separate stacks: one with all cells shifted down (push), kept the same (no-op), and shifted up (pop). The new stack is an element-wise interpolation ("superposition") of these three stacks. The stack elements are vectors, and $a_t = (\mathbf{a}_t, \mathbf{v}_t)$, where the vector $\mathbf{a}_t$ is a probability distribution over the three stack operations. The push operation pushes vector $\mathbf{v}_t$, which can be learned or set to $\mathbf{h}_t$. The stack reading is the top vector element.

## 2.2 RENORMALIZING NONDETERMINISTIC STACK RNN

The main focus of this paper is the renormalizing nondeterministic stack RNN (RNS-RNN) (DuSell & Chiang, 2022). The RNS-RNN's stack module is a simulation of a real-time weighted PDA, complete with its own finite state machine and stack. Let $Q$ be the set of states and $\Gamma$ be the stack alphabet of the PDA. The initial PDA state is $q_0 \in Q$, and the initial stack is $\perp \in \Gamma$. The PDA's computation is governed by weighted *transitions* that manipulate its state and stack contents; a valid sequence of transitions is called a *run*. The PDA is *nondeterministic* in that all possible runs are simulated in parallel. Each run has a weight, which is the product of the weights of its transitions.

The ACTIONS emitted by the controller are the PDA's transition weights. The weights at $t$, denoted $\Delta[t]$, are computed as $\Delta[t] = \exp(\mathbf{W}_a \mathbf{h}_t + \mathbf{b}_a)$. Let $q, r \in Q$ and $u, v \in \Gamma^*$, and let $\Delta[t][q, u \to r, v]$ denote the weight, at timestep $t$, of popping $u$ from the stack, pushing $v$, and transitioning to state $r$ if the previous state was $q$ and the previous stack top was $u$. Transitions have one of three forms, where $x, y \in \Gamma$: $\Delta[t][q, x \to r, xy]$ (push $y$), $\Delta[t][q, x \to r, y]$ (replace $x$ with $y$), and $\Delta[t][q, x \to r, \varepsilon]$ (pop $x$). We say that transitions limited to these three forms are in *restricted form*.

The READING is the marginal distribution, over all runs, of each pair $(r, y) \in Q \times \Gamma$, where $r$ is the current PDA state, and $y$ is the top stack symbol. Let $\tau_i$ be a PDA transition, let $\pi = \tau_1 \cdots \tau_t$ be a PDA run, and let $\psi(\pi) = \prod_{i=1}^{t} \Delta[i][\tau_i]$ be the weight of run $\pi$. Let $\pi \rightsquigarrow t, r, y$ mean that run $\pi$ ends at timestep $t$ in state $r$ with $y$ on top of the stack. The stack reading $\mathbf{r}_t \in \mathbb{R}^{|Q| \cdot |\Gamma|}$ is defined as

$$\mathbf{r}_t[(r, y)] = \frac{\sum_{\pi \rightsquigarrow t, r, y} \psi(\pi)}{\sum_{r', y'} \sum_{\pi \rightsquigarrow t, r', y'} \psi(\pi)}. \tag{1}$$

Equation (1) is sufficient for describing the RNS-RNN mathematically, but it sums over an exponential number of runs, so the RNS-RNN relies on a dynamic programming algorithm to compute it in $O(n^3)$ time (Lang, 1974). The rest of this section describes this algorithm and may safely be skipped unless the reader is interested in these implementation details or in Eqs. (9) to (11).

The algorithm uses a tensor of weights called the *stack WFA*, so named because it can be viewed as a weighted finite automaton (WFA) that encodes the weighted language of all possible stacks the PDA can have at time $t$. Each WFA state is of the form $(i, q, x)$, representing a configuration where the PDA is in state $q$ with stack top $x$ at time $i$. For any WFA transition from $(i, q, x)$ to $(t, r, y)$, its weight is equal to the sum of the weights of all runs that bring the PDA from configuration $(i, q, x)$ to $(t, r, y)$ (possibly over multiple timesteps) without modifying $x$, with the net effect of putting a single $y$ on top of it. The tensor containing the stack WFA's transition weights, also called *inner weights*, is denoted $\gamma$, and elements are written as $\gamma[i \to t][q, x \to r, y]$. For $1 \le t \le n - 1$ and $-1 \le i \le t - 1$,

$$\gamma[-1 \to 0][q, x \to r, y] = \mathbb{I}[q = q_0 \wedge x = \perp \wedge r = q_0 \wedge y = \perp] \qquad \text{init.} \tag{2}$$

$$\gamma[i \to t][q, x \to r, y] = \mathbb{I}[i = t{-}1] \, \Delta[t][q, x \to r, xy] \qquad\qquad\qquad \text{push} \tag{3}$$

$$+ \sum_{s,z} \gamma[i \to t{-}1][q, x \to s, z] \, \Delta[t][s, z \to r, y] \qquad \text{repl.}$$

$$+ \sum_{k=i+1}^{t-2} \sum_{u} \gamma[i \to k][q, x \to u, y] \, \gamma'[k \to t][u, y \to r] \quad \text{pop}$$

$$\gamma'[k \to t][u, y \to r] = \sum_{s,z} \gamma[k \to t{-}1][u, y \to s, z] \, \Delta[t][s, z \to r, \varepsilon] \quad (0 \le k \le t - 2). \tag{4}$$

The RNS-RNN sums over all runs using a tensor of *forward weights* denoted $\alpha$, where the element $\alpha[t][r, y]$ is the total weight of reaching the stack WFA state $(t, r, y)$. These weights are normalized

to get the final stack reading at $t$.

$$\alpha[-1][r, y] = \mathbb{I}[r = q_0 \wedge y = \bot] \tag{5}$$

$$\alpha[t][r, y] = \sum_{i=-1}^{t-1} \sum_{q,x} \alpha[i][q, x]\, \gamma[i \to t][q, x \to r, y] \quad (0 \le t \le n - 1) \tag{6}$$

$$\mathbf{r}_t[(r, y)] = \frac{\alpha[t][r, y]}{\sum_{r', y'} \alpha[t][r', y']}. \tag{7}$$

We have departed slightly from the original definitions of $\gamma$ and $\alpha$ (DuSell & Chiang, 2022), for two reasons: (1) to implement an asymptotic speedup by a factor of $|Q|$ (Butoi et al., 2022), and (2) to fix a peculiarity with the behavior of the initial $\bot$. See Appendix A for details.

## 3 RECOGNITION POWER

In this section, we investigate the power of RNS-RNNs as language recognition devices, proving that they can recognize all CFLs and all intersections of CFLs. These results hold true even when the RNS-RNN is run in real time (one timestep per input, with one extra timestep to read EOS). Although Siegelmann & Sontag (1992) showed that even simple RNNs are as powerful as Turing machines, this result relies on assumptions of infinite precision and unlimited extra timesteps, which generally do not hold true in practice. The same limitation applies to the neural Turing machine (Graves et al., 2014), which, when implemented with finite precision, is no more powerful than a finite automaton, as its tape does not grow with input length. Previously, Stogin et al. (2020) showed that a variant of the superposition stack is at least as powerful as real-time DPDAs. Here we show that RNS-RNNs recognize a much larger superset of languages.

For this section only, we allow parameters to have values of $\pm\infty$, to enable the controller to emit probabilities of exactly zero. Because we use RNS-RNNs here for accepting or rejecting strings (whereas in the rest of the paper, we only use them for predicting the next symbol), we start by providing a formal definition of language recognition for RNNs (cf. Chen et al., 2018).

**Definition 1.** Let $N$ be an RNN controller, possibly augmented with one of the stack modules above. Let $\mathbf{h}_t \in \mathbb{R}^d$ be the hidden state of $N$ after reading $t$ symbols, and let $\sigma$ be the logistic sigmoid function. We say that $N$ *recognizes* language $L$ if there is an MLP layer $y = \sigma(\mathbf{W}_2\, \sigma(\mathbf{W}_1 \mathbf{h}_{|w|+1} + \mathbf{b}_1) + \mathbf{b}_2)$ such that, after reading $w \cdot$ EOS, we have $y > \frac{1}{2}$ iff $w \in L$.

The question of whether RNS-RNNs can recognize all CFLs can be reduced to the question of whether real-time PDAs with transitions in restricted form (Section 2.2) can recognize all CFLs. The real-time requirement does not reduce the power of PDAs (Greibach, 1965), but what about restricted form? We prove that it does not either, and so RNS-RNNs can recognize all CFLs.

**Proposition 1.** *For every context-free language $L$, there exists an RNS-RNN that recognizes $L$.*

*Proof sketch.* We construct a CFG for $L$ and convert it into a modified Greibach normal form, called 2-GNF, which we can convert into a PDA $P$ whose transitions are in restricted form. Then we construct an RNS-RNN that emits, at every timestep, weight 1 for transitions of $P$ and 0 for all others. Then $y > \frac{1}{2}$ iff the PDA ends in an accept configuration. See Appendix B.1 for details. $\quad\square$

**Proposition 2.** *For every finite set of context-free languages $L_1, \ldots, L_k$ over the same alphabet $\Sigma$, there exists an RNS-RNN that recognizes $L_1 \cap \cdots \cap L_k$.*

*Proof sketch.* Without loss of generality, assume $k = 2$. Let $P_1$ and $P_2$ be PDAs recognizing $L_1$ and $L_2$, respectively. We construct a PDA $P$ that uses nondeterminism to simulate $P_1$ *or* $P_2$, but the controller can query $P_1$ and $P_2$'s configurations, and it can set $y > \frac{1}{2}$ iff $P_1$ *and* $P_2$ both end in accept configurations. See Appendix B.2 for details. $\quad\square$

Since the class of languages formed by the intersection of $k$ CFLs is a proper superset of the class formed by the intersection of $(k - 1)$ CFLs (Liu & Weiner, 1973), this means that RNS-RNNs are considerably more powerful than nondeterministic PDAs.

## 4 NON-CONTEXT-FREE LANGUAGES

We now explore the ability of stack RNNs to recognize non-context-free phenomena with a language modeling task on several non-CFLs. Each non-CFL, which we describe below, can be recognized by a real-time three-stack automaton (see Appendix C for details). We also include additional non-CFLs in Appendix C.

$w\#w^R\#w$ The language $\{w\#w^R\#w \mid w \in \{0,1\}^*\}$.

$w\#w$ The language $\{w\#w \mid w \in \{0,1\}^*\}$.

$ww'$ The language $\{ww' \mid w \in \{0,1\}^* \text{ and } w' = \phi(w)\}$, where $\phi$ is the homomorphism $\phi(0) = 2, \phi(1) = 3$.

$ww$ The language $\{ww \mid w \in \{0,1\}^*\}$.

The above languages all include patterns like $w \cdots w$, which are known in linguistics as *cross-serial dependencies*. In Swiss German (Shieber, 1985), the two $w$'s are distinguished by part-of-speech (a sequence of nouns and verbs, respectively), analogous to $ww'$. In Bambara (Culy, 1985), the two $w$'s are the same, but separated by a morpheme $o$, analogous to $w\#w$.

We follow our previous experimental framework (DuSell & Chiang, 2022). If $L$ is a language, let $L_\ell$ be the set of all strings in $L$ of length $\ell$. To sample a string $w \in L$, we first uniformly sample a length $\ell$ from $[40, 80]$, then sample uniformly from $L_\ell$ (we avoid sampling lengths for which $L_\ell$ is empty). So, the distribution from which $w$ is sampled is

$$p_L(w) = \frac{1}{\left|\{\ell \in [40, 80] \mid L_\ell \neq \emptyset\}\right|} \frac{1}{|L_{|w|}|}.$$

We require models to predict an EOS symbol at the end of each string, so each language model $M$ defines a probability distribution $p_M(w)$. Let the per-symbol cross-entropy of a probability distribution $p$ on a set of strings $S$, measured in nats, be defined as

$$H(S, p) = \frac{-\sum_{w \in S} \log p(w)}{\sum_{w \in S}(|w| + 1)}.$$

The $+1$ in the denominator accounts for the fact that the model must predict EOS. Since we know the exact distribution from which the data is sampled (for each non-CFL above, $|L_{|w|}|$ can be computed directly from $|w|$), we can evaluate model $M$ by measuring the *cross-entropy difference* between the learned and true distributions, or $H(S, p_M) - H(S, p_L)$. Lower is better, and 0 is optimal.

We compare five architectures, each of which consists of an LSTM controller connected to a different type of data structure. We include a bare LSTM baseline ("LSTM"). We also include a model that pushes learned vectors of size 10 to a superposition stack ("Sup. 10"), and another that pushes the controller's hidden state ("Sup. h"). Since each of these languages can be recognized by a three-stack automaton, we also tested a model that is connected to three independent instances of the superposition stack, each of which has vectors of size 3 ("Sup. 3-3-3"). Finally, we include an RNS-RNN with $|Q| = 3$ and $|\Gamma| = 3$, where $|Q| = 3$ is sufficient for the model to be able to simulate at least three different stacks. In all cases, the LSTM controller has one layer and 20 hidden units. We encoded all input symbols as one-hot vectors.

Before each training run, we sampled a training set of 10,000 examples and a validation set of 1,000 examples from $p_L$. For each language and architecture, we trained 10 models and report results for the model with the lowest cross-entropy difference on the validation set. For each language, we sampled a single test set that was reused across all training runs. Examples in the test set vary in length from 40 to 100, with 100 examples sampled uniformly from $L_\ell$ for each length $\ell$. Additional training details can be found in Appendix D.

We show the cross-entropy difference on the validation and test sets in Fig. 2. We show results for additional non-CFLs in Appendix E. Strings in $w\#w^R\#w$ contain two hints to facilitate learning: explicit boundaries for $w$, and extra timesteps in the middle, which simplify the task of transferring symbols between stacks (for details, compare the solutions for $w\#w^R\#w$ and $w\#w$ described in Appendix C). Only models that can simulate multiple stacks, Sup. 3-3-3 and RNS 3-3, achieved

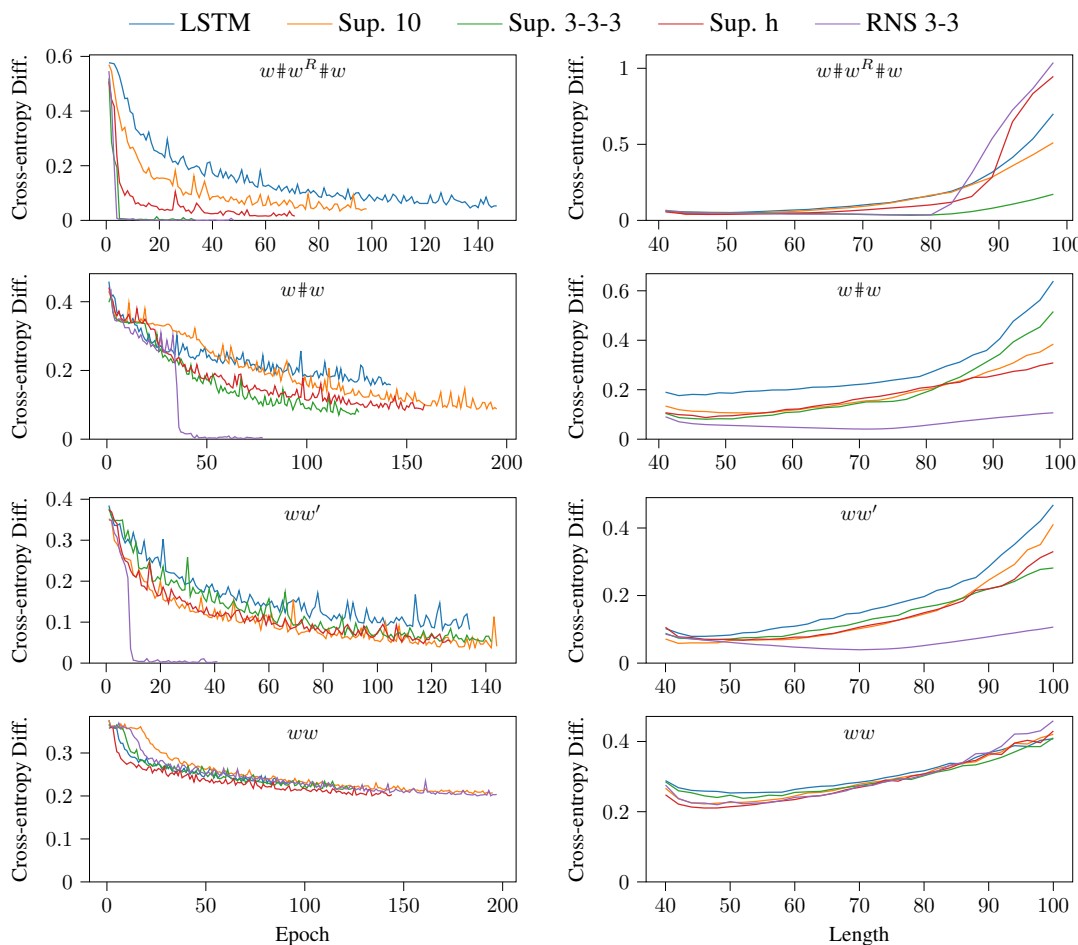

Figure 2: Performance on non-context-free languages. Cross-entropy difference in nats, on the validation set by epoch (left), and on the test set by string length (right). Each line is the best of 10 runs, selected by validation perfomance. On $w\#w^R\#w$, which has both explicit boundaries and extra timesteps in the middle, only multi-stack models (Sup. 3-3-3 and RNS 3-3) achieved optimal cross-entropy. On $w\#w$ and $ww'$, which have no extra timesteps, only RNS 3-3 did.

optimal cross-entropy here, although RNS 3-3 did not generalize well on lengths not seen in training. Only RNS 3-3 effectively learned $w\#w$ and $ww'$. This suggests that although multiple deterministic stacks are enough to learn to copy a string given enough hints in the input (explicit boundaries, extra timesteps for computation), only the nondeterministic stack succeeds when the number of hints is reduced (no extra timesteps). No stack models learned to copy a string without explicit boundaries ($ww$), suggesting a nondeterministic multi-stack model (as opposed to one that uses nondeterminism to simulate multiple stacks) may be needed for such patterns.

## 5 CAPACITY

In this section, we examine how much information each model can transmit through its stack. Consider the language $\{w\#w^R \mid w \in \Sigma^*\}$. It can be recognized by a real-time PDA with $|\Gamma| = 3$ when $|\Sigma| = 2$, but not when $|\Sigma| > 2$, as there is always a sufficiently long $w$ such that there are more possible $w$'s than possible PDA configurations upon reading the $\#$. Similarly, because all the neural stack models we consider here are also real-time, we expect that they will be unable to model context-free languages with sufficiently large alphabets. This is especially relevant to natural languages, which have very large vocabulary sizes.

Neural networks can encode large sets of distinct types in compact vector representations; in fact, Schwartz et al. (2018) and Peng et al. (2018) showed connections between the vector representations in RNNs and the states of WFAs. However, since the RNS-RNN simulates a *discrete* stack, it might struggle on tasks that require it to store strings over alphabets with sizes greater than $|\Gamma|$. On the other hand, a model that uses a stack of *vectors*, like the superposition stack, might model such languages more easily by representing each symbol type as a different cluster of points in a vector space. Here, we make the surprising finding that the RNS-RNN can vastly outperform the superposition stack even for large alphabets, though not always. In addition, to see if we can combine the benefits of nondeterminism with the benefits of vector representations, we propose a new variant of the RNS-RNN that models a PDA with a stack of discrete symbols augmented with vectors.

## 5.1 VECTOR RNS-RNN

The RNS-RNN is computationally expensive for large $|\Gamma|$. To address this shortcoming, we propose a new variant, the Vector RNS-RNN (VRNS-RNN), that uses a stack whose elements are symbols drawn from $\Gamma$ and augmented with *vectors* of size $m$, which it can use to encode large alphabets. Its time and space complexity scale only linearly with $m$. Now, each PDA run involves a stack of elements in $\Gamma \times \mathbb{R}^m$. We assume the initial stack consists of the element $(\bot, \mathbf{v}_0)$, where $\mathbf{v}_0 = \sigma(\mathbf{w}_v)$, and $\mathbf{w}_v$ is a learned parameter. The stack operations now have the following semantics:

**Push** $q, x \to r, y$ If $q$ is the current state and $(x, \mathbf{u})$ is on top of the stack, go to state $r$ and push $(y, \mathbf{v}_t)$ with weight $\Delta[t][q, x \to r, xy]$, where $\mathbf{v}_t = \sigma(\mathbf{W}_v \mathbf{h}_t + \mathbf{b}_v)$.

**Replace** $q, x \to r, y$ If $q$ is the current state and $(x, \mathbf{u})$ is on top of the stack, go to state $r$ and replace $(x, \mathbf{u})$ with $(y, \mathbf{u})$ with weight $\Delta[t][q, x \to r, y]$. Note that we do *not* replace $\mathbf{u}$ with $\mathbf{v}_t$; we replace the discrete symbol only and keep the vector the same. When $x = y$, this is a no-op.

**Pop** $q, x \to r$ If $q$ is the current state and $(x, \mathbf{u})$ is on top of the stack, go to state $r$ and remove $(x, \mathbf{u})$ with weight $\Delta[t][q, x \to r, \varepsilon]$, uncovering the stack element beneath.

Let $\mathbf{v}(\pi)$ denote the top stack vector at the end of run $\pi$. The stack reading $\mathbf{r}_t \in \mathbb{R}^{|Q| \cdot |\Gamma| \cdot m}$ now includes, for each $(r, y) \in Q \times \Gamma$, an interpolation of $\mathbf{v}(\pi)$ for every run $\pi \rightsquigarrow t, r, y$, normalized by the weight of all runs.

$$\mathbf{r}_t[(r, y, j)] = \frac{\sum_{\pi \rightsquigarrow t, r, y} \psi(\pi)\, \mathbf{v}(\pi)[j]}{\sum_{r', y'} \sum_{\pi \rightsquigarrow t, r', y'} \psi(\pi)} \tag{8}$$

We compute the denominator using $\gamma$ and $\alpha$ as before. To compute the numerator, we compute a new tensor $\boldsymbol{\zeta}$ which stores $\psi(\pi)\, \mathbf{v}(\pi)$. For $1 \le t \le n - 1$ and $-1 \le i \le t - 1$,

$$\boldsymbol{\zeta}[-1 \to 0][q, x \to r, y] = \mathbb{I}[q = q_0 \wedge x = \bot \wedge r = q_0 \wedge y = \bot]\, \mathbf{v}_0 \qquad \text{init.}^2 \tag{9}$$

$$\boldsymbol{\zeta}[i \to t][q, x \to r, y] = \mathbb{I}[i = t-1]\, \Delta[t][q, x \to r, xy]\, \mathbf{v}_t \qquad \text{push} \tag{10}$$

$$+ \sum_{s, z} \boldsymbol{\zeta}[i \to t-1][q, x \to s, z]\, \Delta[t][s, z \to r, y] \qquad \text{repl.}$$

$$+ \sum_{k=i+1}^{t-2} \sum_{u} \boldsymbol{\zeta}[i \to k][q, x \to u, y]\, \gamma'[k \to t][u, y \to r]. \qquad \text{pop}$$

We compute $\alpha$ as before, and we compute the normalized stack reading $\mathbf{r}_t$ as follows.

$$\mathbf{r}_t[(r, y, j)] = \frac{\boldsymbol{\eta}_t[r, y][j]}{\sum_{r', y'} \alpha[t][r', y']} \qquad \boldsymbol{\eta}_t[r, y] = \sum_{i=-1}^{t-1} \sum_{q, x} \alpha[i][q, x]\, \boldsymbol{\zeta}[i \to t][q, x \to r, y] \tag{11}$$

## 5.2 EXPERIMENTS

We evaluate models using cross-entropy difference as in Section 4. We express each language $L$ as a PCFG, using the same PCFG definitions as in prior work (DuSell & Chiang, 2020), but modified

---

[2]Our code and experiments implement $\boldsymbol{\zeta}[-1 \to 0][q, x \to r, y] = \mathbf{v}_0$ instead due to a mistake found late in the publication process. Consequently, in Eq. (11), runs can start with *any* $(r, y) \in Q \times \Gamma$ in the numerator, but only $(q_0, \bot)$ in the denominator. Empirically the VRNS-RNN still appears to work as expected.

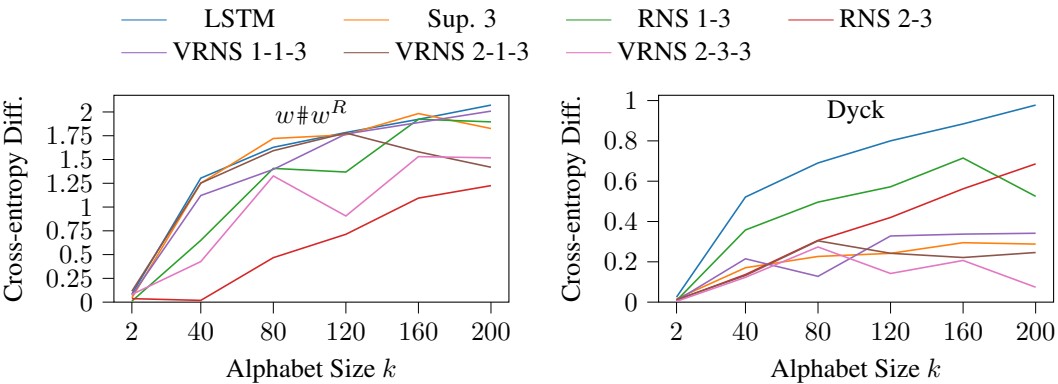

Figure 3: Mean cross-entropy difference on the validation set vs. input alphabet size. Contrary to expectation, RNS 2-3, which models a discrete stack of only 3 symbol types, learns to solve $w\#w^R$ with 200 symbol types more reliably than models with stacks of vectors. On the more complicated Dyck language, vector stacks perform best, with our newly proposed VRNS-RNN performing best.

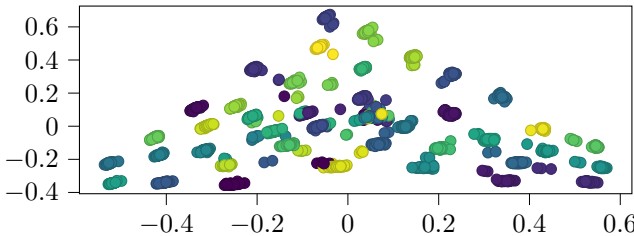

Figure 4: When RNS 2-3 is run on $w\#w^R$ with 40 symbol types, the stack readings are as visualized above. The readings are 6-dimensional vectors, projected down to 2 dimensions using PCA. The color of each point represents the top stack symbol. Points corresponding to the same symbol type cluster together, indicating the RNS-RNN has learned to encode symbols as points in the 5-dimensional simplex. The disorganized points in the middle are from the first and last timesteps of the second half of the string, which appear to be irrelevant for prediction.

to include $k$ symbol types. In order to sample from and compute $p_L(w)$, we used the same sampling and parsing techniques as before (DuSell & Chiang, 2020). We tested the information capacity of each model on two DCFLs, varying their alphabet size $k$ from very small to very large.

$\boldsymbol{w\#w^R}$  The language $\{w\#w^R \mid w \in \{0, 1, \cdots, k-1\}^*\}$.

**Dyck**  The language of strings over the alphabet $\{\,(_1, )_1, (_2, )_2, \cdots, (_k, )_k\,\}$ where all brackets are properly balanced and nested in pairs of $(_i$ and $)_i$.

We compare four types of architecture, including an LSTM baseline ("LSTM"), and a superposition stack that pushes learned vectors of size 3 ("Sup. 3"). We use the notation "RNS $|Q|$-$|\Gamma|$" for RNS-RNNs, and "VRNS $|Q|$-$|\Gamma|$-$m$" for VRNS-RNNs. All details of the controller and training procedure are the same as in Section 4. We varied the alphabet size $k$ from 2 to 200 in increments of 40. For each task, architecture, and alphabet size, we ran 10 random restarts.

In Fig. 3, for each task, we show the mean cross-entropy difference on the validation set as a function of alphabet size; we provide plots of the best performance in Appendix F. On $w\#w^R$, the single-state RNS 1-3, and even VRNS 1-1-3 and Sup. 3, struggled for large $k$. Only the multi-state models RNS 2-3, VRNS 2-1-3, and VRNS 2-3-3 show a clear advantage over the LSTM. Surprisingly, RNS 2-3, which models a discrete stack alphabet of only size 3, attained the best performance on large alphabets; in Fig. 7, it is the only model capable of achieving optimal cross-entropy on all alphabet sizes. On the Dyck language, a more complicated DCFL, the model rankings are as expected: vector stacks (Sup. 3 and VRNS) performed best, with the largest VRNS model performing best. RNS-RNNs still show a clear advantage over the LSTM, but not as much as vector stack RNNs.

Table 1: Validation and test perplexity on the Penn Treebank of the best of 10 random restarts for each architecture. The model with the best test perplexity is our new VRNS-RNN when it combines a modest amount of nondeterminism (3 states and 3 stack symbols) with vectors of size 5.

| Model | Val. ↓ | Test ↓ |
|---|---|---|
| LSTM, 256 units | 129.99 | 125.90 |
| Sup. (push hidden), 247 units | 124.99 | 121.05 |
| Sup. (push learned), $|\mathbf{v}_t| = 22$ | 125.68 | 120.74 |
| RNS 1-29 | 131.17 | 128.11 |
| RNS 2-13 | 128.97 | 122.76 |
| RNS 4-5 | 126.06 | 120.19 |
| VRNS 1-1-256 | 130.60 | 126.70 |
| VRNS 1-1-32 | **124.49** | 120.45 |
| VRNS 1-5-20 | 128.35 | 124.63 |
| VRNS 2-3-10 | 129.30 | 124.03 |
| VRNS 3-3-5 | 124.71 | **120.12** |

If RNS 2-3 has only 3 symbol types at its disposal, how can it succeed on $w \# w^R$ for large $k$? Recall that $\mathbf{r}_t$ is a vector that represents a probability distribution over $Q \times \Gamma$. Perhaps the RNS-RNN, via $\mathbf{r}_t$, represents symbol types as different clusters of points in $\mathbb{R}^{|Q| \cdot |\Gamma|}$. To test this hypothesis, we selected the RNS 2-3 model with the best validation performance on $w \# w^R$ for $k = 40$ and evaluated it on 100 samples drawn from $p_L$. For each symbol between # and EOS, we extracted the stack reading vector computed just prior to predicting that symbol. Aggregating over all 100 samples, we reduced the stack readings to 2 dimensions using principal component analysis. We plot them in Fig. 4, labeling each point according to the symbol type to be predicted just after the corresponding stack reading. Indeed, we see that stack readings corresponding to the same symbol cluster together, suggesting that the model is orchestrating the weights of different runs in a way that causes the stack reading to encode different symbol types as points in the 5-dimensional simplex. We show heatmaps of actual reading vectors in Appendix G.

## 6 NATURAL LANGUAGE MODELING

We now examine how stack RNNs fare on natural language modeling, as the combination of nondeterminism and vector representations in the VRNS-RNN may prove beneficial. Following our prior work (DuSell & Chiang, 2022), we report perplexity on the Penn Treebank as preprocessed by Mikolov et al. (2011). We used the same LSTM and superposition stack baselines, and various sizes of RNS-RNN and VRNS-RNN. The controller has one layer and, unless otherwise noted, 256 hidden units. For each architecture, we trained 10 random restarts and report results for the model with the best validation perplexity. Appendix H has additional details.

We show results in Table 1. Most stack RNNs achieved better test perplexity than the LSTM baseline. The best models are those that simulate more nondeterminism (VRNS when $|Q| = 3$ and $|\Gamma| = 3$, and RNS when $|Q| = 4$ and $|\Gamma| = 5$). Although the superposition stack RNNs outperformed the LSTM baseline, it is the combination of both nondeterminism and vector embeddings (VRNS 3-3-5) that achieved the best performance, combining the ability to process syntax nondeterministically with the ability to pack lexical information into a vector space on the stack.

## 7 CONCLUSION

We showed that the RNS-RNN (DuSell & Chiang, 2022) can recognize all CFLs and a large class of non-CFLs, and it can even learn cross-serial dependencies provided the boundary is explicitly marked, unlike a deterministic multi-stack architecture. We also showed that the RNS-RNN can far exceed the amount of information it seemingly should be able to encode in its stack given its finite stack alphabet. Our newly proposed VRNS-RNN combines the benefits of nondeterminism and vector embeddings, and we showed that it has better performance than other stack RNNs on the Dyck language and a natural language modeling benchmark.

## REPRODUCIBILITY STATEMENT

To facilitate reproducibility, we have publicly released all code we used to conduct our experiments and generate the figures and tables in this paper. During both development and experimentation, we ran our code in containers to simplify reproducing our software environment. Our code includes the original Docker image definition we used, as well as the exact shell commands we used for each experiment, figure, and table. We have thoroughly documented our experimental settings in Sections 4 to 6 and Appendices D and H.

## ACKNOWLEDGMENTS

This research was supported in part by a Google Faculty Research Award to Chiang. We would like to thank Darcey Riley and Stephen Bothwell for their comments on an earlier draft of this paper, and the Center for Research Computing at the University of Notre Dame for providing the computing infrastructure for our experiments.

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

## A  CHANGES TO THE RNS-RNN

In this section, we describe two minor changes to the original definition of the RNS-RNN that improve its time complexity and expressivity. We first reiterate the original definitions for $\gamma$ and $\alpha$

(DuSell & Chiang, 2022). For $0 \le i < t \le n-1$,

$$
\begin{aligned}
\gamma[i \to t][q, x \to r, y] = & \\
& \mathbb{I}[i = t{-}1] \; \Delta[t][q, x \to r, xy] & \text{push} \\
& + \sum_{s,z} \gamma[i \to t{-}1][q, x \to s, z] \; \Delta[t][s, z \to r, y] & \text{repl.} \\
& + \sum_{k=i+1}^{t-2} \sum_u \sum_{s,z} \gamma[i \to k][q, x \to u, y] \; \gamma[k \to t{-}1][u, y \to s, z] \; \Delta[t][s, z \to r, \varepsilon] & \text{pop}
\end{aligned}
$$

(12)

$$
\alpha[0][r, y] = \mathbb{I}[r = q_0 \wedge y = \bot] \tag{13}
$$

$$
\alpha[t][r, y] = \sum_{i=0}^{t-1} \sum_{q,x} \alpha[i][q, x] \, \gamma[i \to t][q, x \to r, y] \qquad (1 \le t \le n). \tag{14}
$$

Note that in an RNN where $\mathbf{r}_{t-1}$ is used to compute $\mathbf{h}_t$ and $\mathbf{y}_t$, the last timestep $t = n$ is not needed, so $\Delta[t]$ is only defined for $1 \le t \le n-1$, and $\gamma$ and $\alpha$ only need to be computed for $t \le n-1$.

Now, we describe the two changes we have made to Eqs. (12) to (14) to arrive at Eqs. (2) to (6).

**Asymptotic speedup**    As can be seen in Eqs. (7) and (12) to (14), the RNS-RNN's time complexity is $O(|Q|^4|\Gamma|^3 n^3)$, and its space complexity is $O(|Q|^2|\Gamma|^2 n^2)$. The computational complexity of the RNS-RNN limits us to relatively small sizes for $Q$ and $\Gamma$. However, it is possible to improve its asymptotic time complexity with respect to $|Q|$ with a simple change: precomputing the product $\gamma[k \to t{-}1][u, y \to s, z] \; \Delta[t][s, z \to r, \varepsilon]$ used in the pop rule in Eq. (12), which we store in a tensor $\gamma'$. This reduces the time complexity of the RNS-RNN to $O(|Q|^3|\Gamma|^3 n^2 + |Q|^3|\Gamma|^2 n^3)$, allowing us to train larger models.

**Bottom symbol fix**    There are some peculiarities of the initial $\bot$ marker that are not described in previous work: (1) the initial instance of $\bot$ at the bottom of the stack can never be popped or replaced, (2) any symbol that sits directly above it cannot be popped (but it can be replaced), and (3) the $\bot$ symbol type can be reused freely elsewhere in the stack. Point (2) is odd because the $\bot$ symbol can never be uncovered again after the first timestep, possibly complicating detection of the bottom of the stack later on. We rectify this by allowing the symbol above the initial $\bot$ to be popped, and allowing the initial $\bot$ symbol to be replaced with a different symbol type at any time. We do this by simulating an extra push action at $t = -1$.

## B    PROOFS OF LANGUAGE RECOGNITION RESULTS

### B.1    PROOF OF PROPOSITION 1

In this section, we prove that the restricted form of real-time PDA used in the RNS-RNN can recognize all CFLs. We then show how to convert PDAs in this form to RNS-RNN recognizers (assuming some parameters can have values of $\pm\infty$), proving that RNS-RNNs can recognize all CFLs.

**Definition 2.** A *pushdown automaton* is a tuple $(Q, \Sigma, \Gamma, \delta, q_0, F, \bot)$, where

- $Q$ is a finite set of states

- $\Sigma$ is a finite input alphabet

- $\Gamma$ is a finite stack alphabet

- $\delta \subseteq Q \times \Gamma \times (\Sigma \cup \varepsilon) \times Q \times \Gamma^*$ is a set of transitions

- $q_0 \in Q$ is the start state

- $F \subseteq Q$ is the set of accept states

- $\perp \in \Gamma$ is the bottom stack symbol.

A PDA always starts in state $q_0$ with a stack consisting of $\perp$, and it accepts its input iff there is a run that terminates in an accept state with $\perp$ on top of the stack.

**Definition 3.** A *restricted* PDA is one whose transitions have one of the following forms, where $q, r \in Q$, $a \in \Sigma$, and $x, y \in \Gamma$:

$$q, x \xrightarrow{a} r, xy \qquad\qquad \text{push } y \text{ on top of } x$$
$$q, x \xrightarrow{a} r, y \qquad\qquad \text{replace } x \text{ with } y$$
$$q, x \xrightarrow{a} r, \varepsilon \qquad\qquad \text{pop } x.$$

The usual construction for removing non-scanning transitions (Autebert et al., 1997) involves converting to Greibach normal form (GNF) and then converting to a PDA. However, this construction produces transitions of the form $q, x \xrightarrow{a} r, zy$ where $x \neq z$, which our restricted form does not allow. Simulating such transitions in a restricted PDA presents a challenge because it requires performing a replace and then a push while scanning only one symbol. To simulate such transitions, we need to use a modified GNF, defined below.

**Lemma 3.** *For any CFG $G$, there is a CFG equivalent to $G$ that has the following form (called 2–Greibach normal form):*

- *The start symbol $S$ does not appear on any right-hand side.*

- *Every rule has one of the following forms:*

$$S \to \varepsilon$$
$$A \to a$$
$$A \to abB_1 \cdots B_p \qquad\qquad p \geq 0.$$

*Proof.* Convert $G$ to Greibach normal form. Then for every rule $A \to aA_1 \cdots A_m$ and every rule $A_1 \to bB_1 \cdots B_\ell$, substitute the second rule into the first to obtain $A \to abB_1 \cdots B_\ell A_2 \cdots A_m$. Then discard the first rule. $\qquad\square$

**Lemma 4.** *For any CFG $G$ in 2-GNF, there is a PDA equivalent to $G$ whose transitions all have one of the forms:*

$$q, x \xrightarrow{a} r, xy_1 \cdots y_k$$
$$q, x \xrightarrow{a} r, y \qquad\qquad\qquad (15)$$
$$q, x \xrightarrow{a} r, \varepsilon.$$

*Proof.* We split the rules into four cases:

$$S \to \varepsilon$$
$$A \to a$$
$$A \to ab$$
$$A \to abB_1 \cdots B_p \qquad p \geq 1.$$

The PDA has an initial state $q_0$ and a main loop state $q_{\text{loop}}$. It works by maintaining all of the unclosed constituents on the stack, which initially is $S$. The state $q_{\text{loop}}$ is an accept state. For now, we allow the PDA to have one non-scanning transition, $q_0, \perp \xrightarrow{\varepsilon} q_{\text{loop}}, \perp S$.

If $G$ has the rule $S \to \varepsilon$, we make state $q_0$ an accept state.

For each rule in $G$ of the form $A \to a$, we add a pop transition $q_{\text{loop}}, A \xrightarrow{a} q_{\text{loop}}, \varepsilon$.

For each rule in $G$ of the form $A \to ab$, we add a new state $q$, a replace transition $q_{\text{loop}}, A \xrightarrow{a} q, A$, and a pop transition $q, A \xrightarrow{b} q_{\text{loop}}, \varepsilon$. This simply scans two symbols while popping $A$.

For each rule in $G$ of the form $A \to abB_1 \cdots B_p$ where $p \geq 1$, we add a new state $q$, a replace transition $q_{\text{loop}}, A \xrightarrow{a} q, B_p$, and a push transition $q, B_p \xrightarrow{b} q_{\text{loop}}, B_p B_{p-1} \cdots B_1$. These two transitions

are equivalent to scanning two symbols while replacing $A$ with $B_p B_{p-1} \cdots B_1$ on the stack. Note that we have taken advantage of the fact that 2-GNF affords us *two* scanned input symbols to work around the restriction that push transitions of the form $q, x \xrightarrow{a} r, x y_1 \cdots y_k$ cannot modify the top symbol and push new symbols in the same step. We have split this action into a replace transition followed by a push transition, using the state machine to remember what to scan and push after the replace transition.

Finally, we remove the non-scanning transition $q_0, \bot \xrightarrow{\varepsilon} q_{\text{loop}}, \bot S$, and for every transition $q_{\text{loop}}, S \xrightarrow{a} r, \alpha$, we add a push transition $q_0, \bot \xrightarrow{a} r, \bot \alpha$. Now all transitions are scanning. $\qquad \square$

**Lemma 5.** *For any PDA $P$ in the form (15), there is a PDA equivalent to $P$ in restricted form.*

*Proof.* At this point, there is a maximum length $k$ such that $q, x \xrightarrow{a} r, x\alpha$ is a transition and $k = |\alpha|$. We redefine the stack alphabet of the PDA to be $\Gamma' = \Gamma \cup \Gamma^2 \cdots \cup \Gamma^k$. Stack symbols now represent strings of the original stack symbols. Let $\boxed{\alpha}$ denote a single stack symbol for any string $\alpha$. We replace every push transition $q, x \xrightarrow{a} r, x\beta$ with push transitions $q, \boxed{\alpha x} \xrightarrow{a} r, \boxed{\alpha x}\boxed{\beta}$ for all $\alpha \in \bigcup_{i=0}^{k-1} \Gamma^i$. We replace every replace transition $q, x \xrightarrow{a} r, y$ with replace transitions $q, \boxed{\alpha x} \xrightarrow{a} r, \boxed{\alpha y}$ for all $\alpha$. And we replace every pop transition $q, x \xrightarrow{a} r, \varepsilon$ with replace transitions $q, \boxed{\alpha x} \xrightarrow{a} r, \boxed{\alpha}$ for all $\alpha$ with $|\alpha| \geq 1$, and a pop transition $q, \boxed{x} \xrightarrow{a} r, \varepsilon$. $\qquad \square$

So for every CFL $L$, there exists a restricted PDA $P$ that recognizes $L$. The last step is to construct an RNS-RNN. As noted in Section 3, here we do allow parameters with values of $\pm\infty$.

**Lemma 6.** *For any restricted-form PDA $P$ that recognizes language $L$, there is an RNS-RNN that recognizes $L$.*

*Proof.* Let $P = (Q, \Sigma, \Gamma, \delta, q_0, F, \bot)$.

We write $ct(w)$ for the total number of runs of $P$ that read $w$ and end with $\bot$ on top of the stack. We assume that $ct(w) > 0$; this can always be ensured by adding to $P$ an extra non-accepting state $q_{\text{trap}}$ and transitions $q_0, \bot \xrightarrow{a} q_{\text{trap}}, \bot$ and $q_{\text{trap}}, \bot \xrightarrow{a} q_{\text{trap}}, \bot$ for all $a \in \Sigma$.

For any state set $X \subseteq Q$, we write $ct(w, X)$ for the total number of runs of $P$ that read $w$ and end in a state in $X$ with $\bot$ on top of the stack. Then for any string $w$, if $w \in L$, then $ct(w, F) \geq 1$; otherwise, $ct(w, F) = 0$.

We use the following definition for the LSTM controller of the RNS-RNN, where $\mathbf{i}_t$, $\mathbf{f}_t$, and $\mathbf{o}_t$ are the input, forget, and output gates, respectively, and $\mathbf{g}_t$ is the candidate memory cell.

$$\mathbf{i}_t = \sigma(\mathbf{W}_i \begin{bmatrix} \mathbf{x}_t \\ \mathbf{h}_{t-1} \end{bmatrix} + \mathbf{b}_i)$$

$$\mathbf{f}_t = \sigma(\mathbf{W}_f \begin{bmatrix} \mathbf{x}_t \\ \mathbf{h}_{t-1} \end{bmatrix} + \mathbf{b}_f)$$

$$\mathbf{g}_t = \tanh(\mathbf{W}_g \begin{bmatrix} \mathbf{x}_t \\ \mathbf{h}_{t-1} \end{bmatrix} + \mathbf{b}_g)$$

$$\mathbf{o}_t = \sigma(\mathbf{W}_o \begin{bmatrix} \mathbf{x}_t \\ \mathbf{h}_{t-1} \end{bmatrix} + \mathbf{b}_o)$$

$$\mathbf{c}_t = \mathbf{f}_t \odot \mathbf{c}_{t-1} + \mathbf{i}_t \odot \mathbf{g}_t$$

$$\mathbf{h}_t = \mathbf{o}_t \odot \tanh(\mathbf{c}_t)$$

Construct an RNS-RNN as follows. At each timestep $t$, upon reading input embedding $\mathbf{x}_t$, make the controller emit a weight of 1 for all transitions of $P$ that scan input symbol $w_t$, and a weight of 0 for all other transitions (by setting the corresponding weights of $\Delta[t]$ to $-\infty$).

Let $n = |w|$. After reading $w$, the stack reading $\mathbf{r}_n$ is the probability distribution of states and top stack symbols of $P$ after reading $w$, which the controller can use to compute $\mathbf{h}_{n+1}$ and the MLP

output $y$ as follows. Let

$$p = \sum_{f \in F} \mathbf{r}_n[(f, \perp)] = \frac{ct(w, F)}{ct(w)}.$$

Note that $p$ is positive if $w \in L$, and zero otherwise. An affine layer connects $\mathbf{h}_n$, $\mathbf{x}_{n+1}$, and $\mathbf{r}_n$ to the candidate memory cell $\mathbf{g}_{n+1}$ (recall that the input at $n+1$ is EOS). Designate a unit $g_{n+1}^{\text{accept}}$ in $\mathbf{g}_{n+1}$. For each $f \in F$, set the incoming weight from $\mathbf{r}_n[(f, \perp)]$ to 1 and all other incoming weights to 0, so that $g_{n+1}^{\text{accept}} = \tanh(p)$. Set this unit's input gate to 1 and forget gate to 0, so that memory cell $c_{n+1}^{\text{accept}} = \tanh(p)$. Set its output gate to 1, so that hidden unit $h_{n+1}^{\text{accept}} = \tanh(\tanh(p))$, which is positive if $w \in L$ and zero otherwise.

Finally, to compute $y$, use one hidden unit $y_1$ in the output MLP layer, and set the incoming weight from $h_{n+1}^{\text{accept}}$ to 1, and all other weights to 0. So $y_1 = \sigma(\tanh(\tanh(p)))$, which is greater than $\frac{1}{2}$ if $w \in L$ and equal to $\frac{1}{2}$ otherwise. Set the weight connecting $y_1$ to $y$ to 1, and set the bias term to $-\frac{1}{2}$, so that $y = \sigma(\sigma(\tanh(\tanh(p))) - \frac{1}{2})$, which is greater than $\frac{1}{2}$ if $w \in L$, and equal to $\frac{1}{2}$ otherwise. □

## B.2 PROOF OF PROPOSITION 2

Without loss of generality, assume $k = 2$. Let $P_1 = (Q_1, \Sigma, \Gamma_1, \delta_1, s_1, F_1, \perp)$ and $P_2 = (Q_2, \Sigma, \Gamma_2, \delta_2, s_2, F_2, \perp)$ be restricted-form PDAs recognizing $L_1$ and $L_2$, respectively. We can construct both so that $Q_1 \cap Q_2 = \emptyset$, and $s_1$ and $s_2$ each have no incoming transitions. Construct a new PDA

$$P = (Q, \Sigma, \Gamma_1 \cup \Gamma_2, \delta, s, F)$$
$$Q = (Q_1 \setminus \{s_1\}) \cup (Q_2 \setminus \{s_2\}) \cup \{s\}$$
$$\delta(q, x, a) = \begin{cases} \delta_1(q, x, a) & q \in Q_1 \setminus \{s_1\} \\ \delta_2(q, x, a) & q \in Q_2 \setminus \{s_2\} \\ \delta_1(s_1, x, a) \cup \delta_2(s_2, x, a) & q = s \end{cases}$$
$$F = \begin{cases} (F_1 \setminus \{s_1\}) \cup (F_2 \setminus \{s_2\}) \cup \{s\} & s_1 \in F_1 \wedge s_2 \in F_2 \\ (F_1 \setminus \{s_1\}) \cup (F_2 \setminus \{s_2\}) & \text{otherwise.} \end{cases}$$

So far, this is just the standard union construction for PDAs.

Construct an RNS-RNN that sets $\Delta[t]$ according to $\delta$, and assume $ct(w) > 0$, as in Lemma 6. Let

$$p_1 = \sum_{f \in F_1} \mathbf{r}_n[(f, \perp)] = \frac{ct(w, F_1)}{ct(w)}$$
$$p_2 = \sum_{f \in F_2} \mathbf{r}_n[(f, \perp)] = \frac{ct(w, F_2)}{ct(w)}.$$

Let $n = |w|$. For each $p_i$, designate a unit $g_{i,n+1}^{\text{accept}}$ in $\mathbf{g}_{n+1}$ that will be positive if $p_1 > 0$ and negative if $p_1 = 0$. In order to ensure that $g_{i,n+1}^{\text{accept}} \neq 0$, we subtract a small value from $p_i$ that is smaller than the smallest possible non-zero value of $p_i$. The RNS-RNN computes this value as follows. Let $b = |Q|(2|\Gamma| + 1)$, which is the maximum number of choices $P$ can make from any given configuration. Designate one memory cell $c_t^{\text{offset}}$ in $\mathbf{c}_t$. At the first timestep, $c_t^{\text{offset}}$ is initialized to $\frac{1}{b}$, and at each subsequent timestep, the forget gate is used to multiply this cell by $\frac{1}{b}$ (the input gate is set to 0). So after reading $t$ symbols, $c_t^{\text{offset}} = \frac{1}{b^t}$. Set the output gate for that cell to 1, so there is a hidden unit $h_n^{\text{offset}}$ in $\mathbf{h}_n$ that contains the value $\tanh(\frac{1}{b^n})$. The smallest non-zero value of $p_i$ is $\frac{1}{ct(w)}$, and since $ct(w) \leq b^n$ and $\tanh(x) < x$ when $x > 0$, $h_n^{\text{offset}}$ is smaller than it.

As for the incoming weights to $g_{i,n+1}^{\text{accept}}$, for each $f \in F_i$, set the weight for $\mathbf{r}_n[(f, \perp)]$ to 1, and set the weight for $h_n^{\text{offset}}$ to $-1$; set all other weights to 0. So $g_{i,n+1}^{\text{accept}} = \tanh(p_i - h_n^{\text{offset}})$, which is positive if $p_i > 0$ and negative otherwise. Set this unit's input gate to 1 and forget gate to 0, so that memory cell $c_{i,n+1}^{\text{accept}} = g_{i,n+1}^{\text{accept}}$. Set its output gate to 1, so that hidden unit $h_{i,n+1}^{\text{accept}} = \tanh(g_{i,n+1}^{\text{accept}})$, which is positive if $p_i > 0$ and negative otherwise.

In the output MLP's hidden layer, for each $h_{i,n+1}^{\text{accept}}$, include a unit $y_i$, and set its incoming weight from $h_{i,n+1}^{\text{accept}}$ to $\infty$, and all other weights to 0. So $y_i = \sigma(\infty \cdot h_{i,n+1}^{\text{accept}})$, which is 1 if $p_i > 0$ and 0 otherwise. Finally, to compute $y$, set the incoming weights from each $y_i$ to 1, and set the bias term to $-\frac{3}{2}$. So $y = \sigma(y_1 + y_2 - \frac{3}{2})$, which is greater than $\frac{1}{2}$ if both $p_1 > 0$ and $p_2 > 0$, meaning $w \in L_1 \cap L_2$, and less than $\frac{1}{2}$ otherwise.

## C  ADDITIONAL DISCUSSION OF NON-CONTEXT-FREE LANGUAGES

Below we discuss each of the non-CFLs of Section 4 in more detail, including details of how a real-time multi-stack automaton could recognize it. We also describe three non-CFLs not included in Section 4.

$\mathbf{a}^n\mathbf{b}^n\mathbf{c}^n$  The language $\{\texttt{a}^n\texttt{b}^n\texttt{c}^n \mid n \geq 0\}$, a classic example of a non-CFL (Sipser, 2013). A two-stack automaton can recognize this language as follows. While reading the $\texttt{a}$'s, push them to stack 1. While reading the $\texttt{b}$'s, match them with $\texttt{a}$'s popped from stack 1 while pushing $\texttt{b}$'s to stack 2. While reading the $\texttt{c}$'s, match them with $\texttt{b}$'s popped from stack 2. As the stacks are only needed to remember the *count* of each symbol type, this language is also an example of a counting language; Weiss et al. (2018) showed that LSTMs can learn this language by using their memory cells as counters.

$\mathbf{w}\#\mathbf{w}^R\#\mathbf{w}$  The language $\{w\#w^R\#w \mid w \in \{0,1\}^*\}$. A two-stack automaton can recognize it as follows. While reading the first $w$, push it to stack 1. While reading the middle $w^R$, match it with symbols popped from stack 1 while pushing $w^R$ to stack 2. While reading the last $w$, match it with symbols popped from stack 2. The explicit $\#$ symbols are meant to make it easier for a model to learn when to transition between these three phases.

$\mathbf{w}\#^n\mathbf{w}$  The language $\{w\#^nw \mid w \in \{0,1\}^*, n \geq 0, \text{ and } |w| = n\}$. A two-stack automaton can recognize it as follows. While reading the first $w$, push it to stack 1. While reading $\#^n$, move the symbols from stack 1 to stack 2 in reverse. While reading the final $w$, match it with symbols popped from stack 2. Unlike $w\#w^R\#w$, the middle $\#^n$ section offers a model few hints that it should push $w$ to the stack beforehand.

$\mathbf{w}\#\mathbf{w}$  The language $\{w\#w \mid w \in \{0,1\}^*\}$. A three-stack automaton can recognize this language as follows. Let $w = uv$ where $|u| = |v|$ (for simplicity assume $|w|$ is even). While reading the first $u$, push it to stack 1. While reading the first $v$, push it to stack 2, and move the symbols from stack 1 to stack 3 in reverse. While reading the second $u$, match it with symbols popped from stack 3, and move the symbols from stack 2 to stack 1 in reverse. While reading the second $v$, match it with symbols popped from stack 1. The explicit $\#$ symbol is meant to make learning this task easier.

$\mathbf{w}\mathbf{w}'$  The language $\{ww' \mid w \in \{0,1\}^* \text{ and } w' = \phi(w)\}$, where $\phi$ is the homomorphism $\phi(0) = 2$, $\phi(1) = 3$. A three-stack automaton can recognize this language using a similar strategy to $w\#w$. In this case, a switch to a different alphabet, rather than a $\#$ symbol, marks the second half of the string.

$\mathbf{w}\mathbf{w}^R\mathbf{w}$  The language $\{ww^Rw \mid w \in \{0,1\}^*\}$. A two-stack automaton can recognize it using a similar strategy to $w\#w^R\#w$, but it must nondeterministicaly guess $|w|$.

$\mathbf{w}\mathbf{w}$  The language $\{ww \mid w \in \{0,1\}^*\}$, another classic example of a non-CFL (Sipser, 2013). A three-stack automaton can recognize it using a similar strategy to $w\#w$, except it must nondeterministically guess $|w|$.

## D  ADDITIONAL DETAILS FOR FORMAL LANGUAGE EXPERIMENTS

Here we describe the training procedure used for the non-CFL and capacity experiments in Sections 4 and 5 in more detail. We trained each model by minimizing its cross-entropy (summed over the timestep dimension of each batch) on the training set, and we used per-symbol cross-entropy on the validation set as the early stopping criterion. We optimized the parameters of the model with Adam. For each training run, we randomly sampled the initial learning rate from a log-uniform distribution over $[5 \times 10^{-4}, 1 \times 10^{-2}]$, and we used a gradient clipping threshold of 5. We initialized all fully-connected layers except for those in the LSTM controller with Xavier uniform initialization,

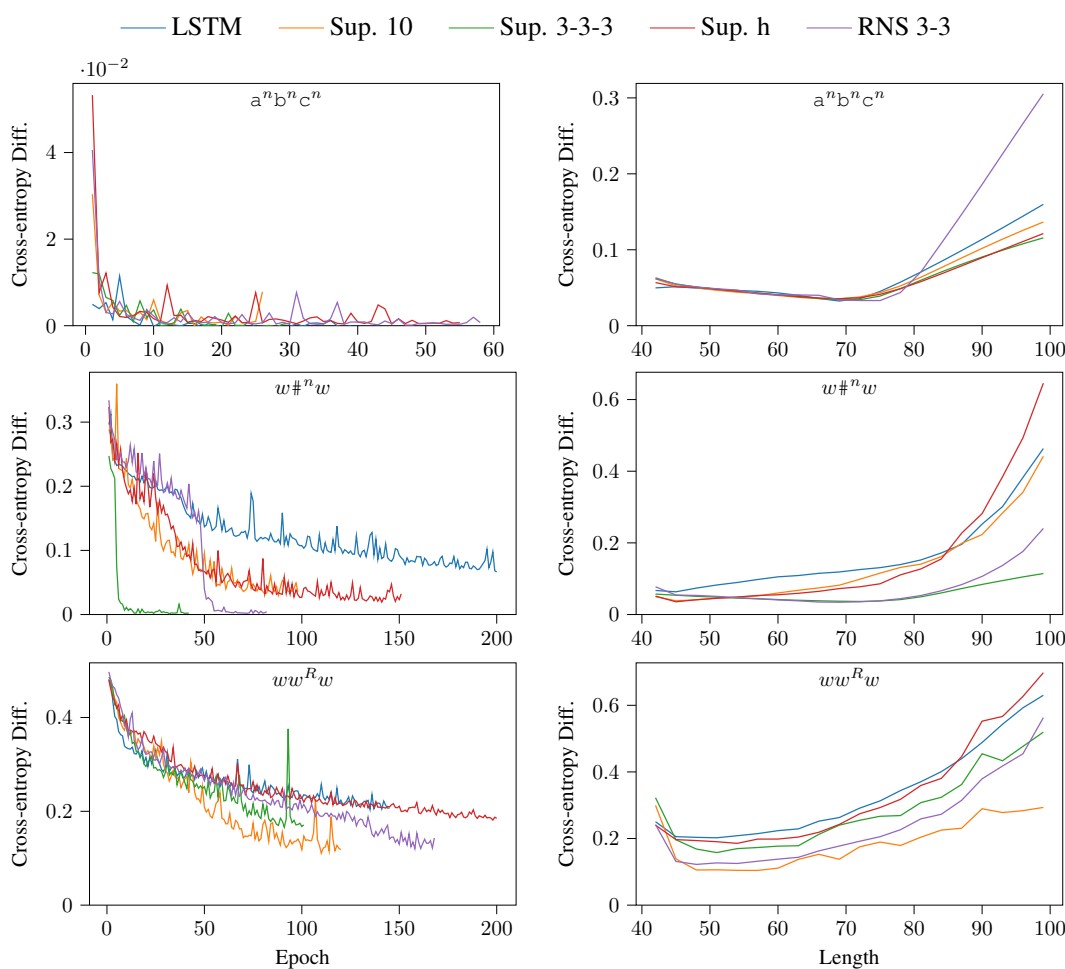

Figure 5: Performance on three non-CFLs not included in Section 4. Cross-entropy difference in nats, on the validation set by epoch (left), and on the test set by string length (right). Each line is the best of 10 runs, selected by validation perfomance. All models easily solved $a^n b^n c^n$. As with $w\#^n w$, only multi-stack models (Sup. 3-3-3 and RNS 3-3) solved $w\#^n w$. As with $ww$, no models solved $ww^R w$.

and all other parameters uniformly from $[-0.1, 0.1]$. We used mini-batches of size 10; each batch always contained examples of equal lengths. We randomly shuffled batches before each epoch. We multiplied the learning rate by 0.9 after 5 epochs of no improvement on the validation set, and we stopped early after 10 epochs of no improvement.

## E ADDITIONAL RESULTS FOR NON-CFL EXPERIMENTS

In Fig. 5, we show results for three non-CFLs which we did not include in Section 4. The input and forget gates of the LSTM controller are not tied, so the values of its memory cells are not bounded to $(0, 1)$ and can be used as counters. All models easily learned $a^n b^n c^n$, likely because the LSTM controller by itself can solve it using a counting mechanism (Weiss et al., 2018). Only the models capable of simulating multiple stacks, Sup. 3-3-3 and RNS 3-3, achieved optimal cross-entropy on $w\#w^R\#w$ and $w\#^n w$. No models succeeded on the unmarked copying tasks ($ww^R w$ and $ww$), although Sup. 10 achieved the best performance on the test set for $ww^R w$.

## F    ADDITIONAL RESULTS FOR CAPACITY EXPERIMENTS

Here, we describe the languages of Section 5 in more detail, plus an additional language, $ww^R$.

$\boldsymbol{w\#w^R}$  The language $\{w\#w^R \mid w \in \{0, 1, \cdots, k-1\}^*\}$. This is a simple deterministic CFL.

**Dyck**  The language of strings over the alphabet $\{\,(_1, )_1, (_2, )_2, \cdots, (_k, )_k\,\}$ where all brackets are properly balanced and nested in pairs of $(_i$ and $)_i$. This is a more complicated but still deterministic CFL.

$\boldsymbol{ww^R}$  The language $\{ww^R \mid w \in \{0, 1, \cdots, k-1\}^*\}$. This is a nondeterministic CFL which requires a model to guess $|w|$.

In Fig. 6, we show the results of the same experiments as in Section 5, but with standard deviations included. We also show results for the language $ww^R$. In Fig. 7, for the same experiments, we show the minimum cross-entropy difference on the validation set out of all 10 random restarts, rather than the mean. None of the models performed significantly better than the LSTM on the nondeterministic $ww^R$ language, suggesting that they cannot simultaneously combine the tricks of encoding symbol types as points in high-dimensional space and nondeterministically guessing $|w|$. Although the VRNS-RNN does combine both nondeterminism and vectors, the nondeterminism only applies to the discrete symbols of $\Gamma$, of which there are no more than 3, far fewer than $k$ when $k \geq 40$.

## G    HEATMAPS OF STACK READING VECTORS

In Fig. 8 we show heatmaps of stack reading vectors across time on an example string in $w\#w^R$ when $k = 40$.

## H    ADDITIONAL DETAILS FOR NATURAL LANGUAGE EXPERIMENTS

Here we describe the Penn Treebank experiments in more detail. For each architecture, we used a word embedding layer of the same size as the hidden state. In order to preserve context across batches, we trained all models using truncated backpropagation through time (BPTT), treating each dataset as one long sequence and limiting batches to a length of 35. As we noted previously (DuSell & Chiang, 2022), training stack RNNs with truncated BPTT requires bounding the size of the stack data structure, as having it grow indefinitely from batch to batch would be computationally infeasible. We limited the depth of the superposition stack to 10, following Yogatama et al. (2018) and our previous paper (DuSell & Chiang, 2022). To limit the size of the RNS-RNN, we used the incremental execution technique we devised previously (DuSell & Chiang, 2022), which limits non-zero entries in $\gamma$ to those where $t - i \leq D$, for some constant window size $D$. We applied the same technique to the VRNS-RNN by imposing the same limitation on both $\gamma$ and $\zeta$, restricting non-zero entries of $\zeta$ to those where $t - i \leq D$. In both cases, we set $D = 35$. We used the standard train/validation/test splits for the Penn Treebank.

We trained each model by minimizing its cross-entropy (averaged over the timestep dimension of each batch) on the training set, using per-symbol perplexity on the validation set as the early stopping criterion. We optimized the parameters of the model with simple SGD. For each training run, we randomly sampled the initial learning rate from a log-uniform distribution over $[1, 100]$, and the gradient clipping threshold from a log-uniform distribution over $[0.0112, 1.12]$. We initialized all parameters uniformly from $[-0.05, 0.05]$. We used a mini-batch size of 32. We divided the learning rate by 1.5 whenever validation perplexity did not improve after an epoch, and we stopped early after 2 epochs of no improvement.

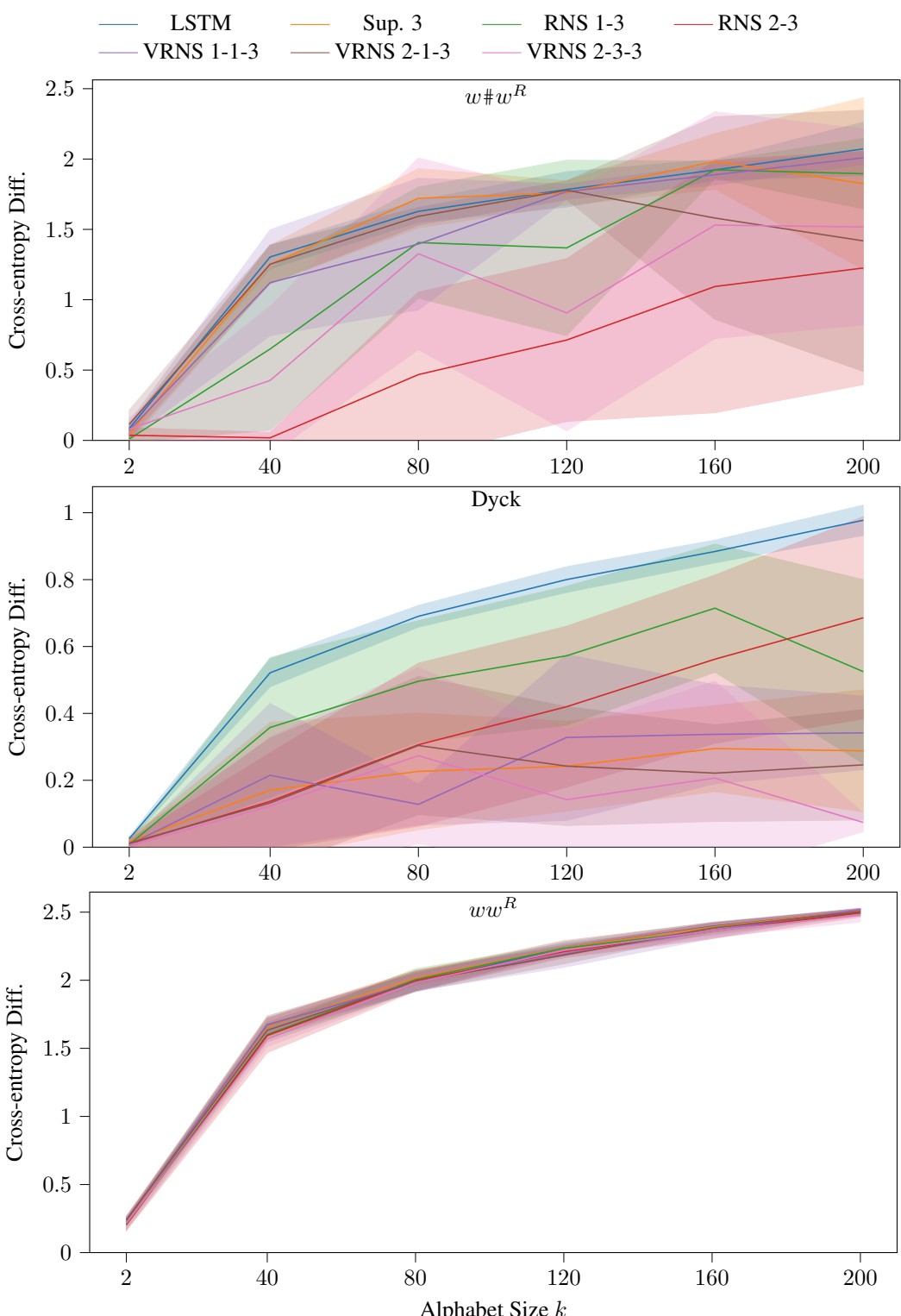

Figure 6: Mean cross-entropy difference on the validation set vs. input alphabet size. The shaded regions indicate one standard deviation. We include experiments on $ww^R$; no models performed significantly better on average than the LSTM baseline.

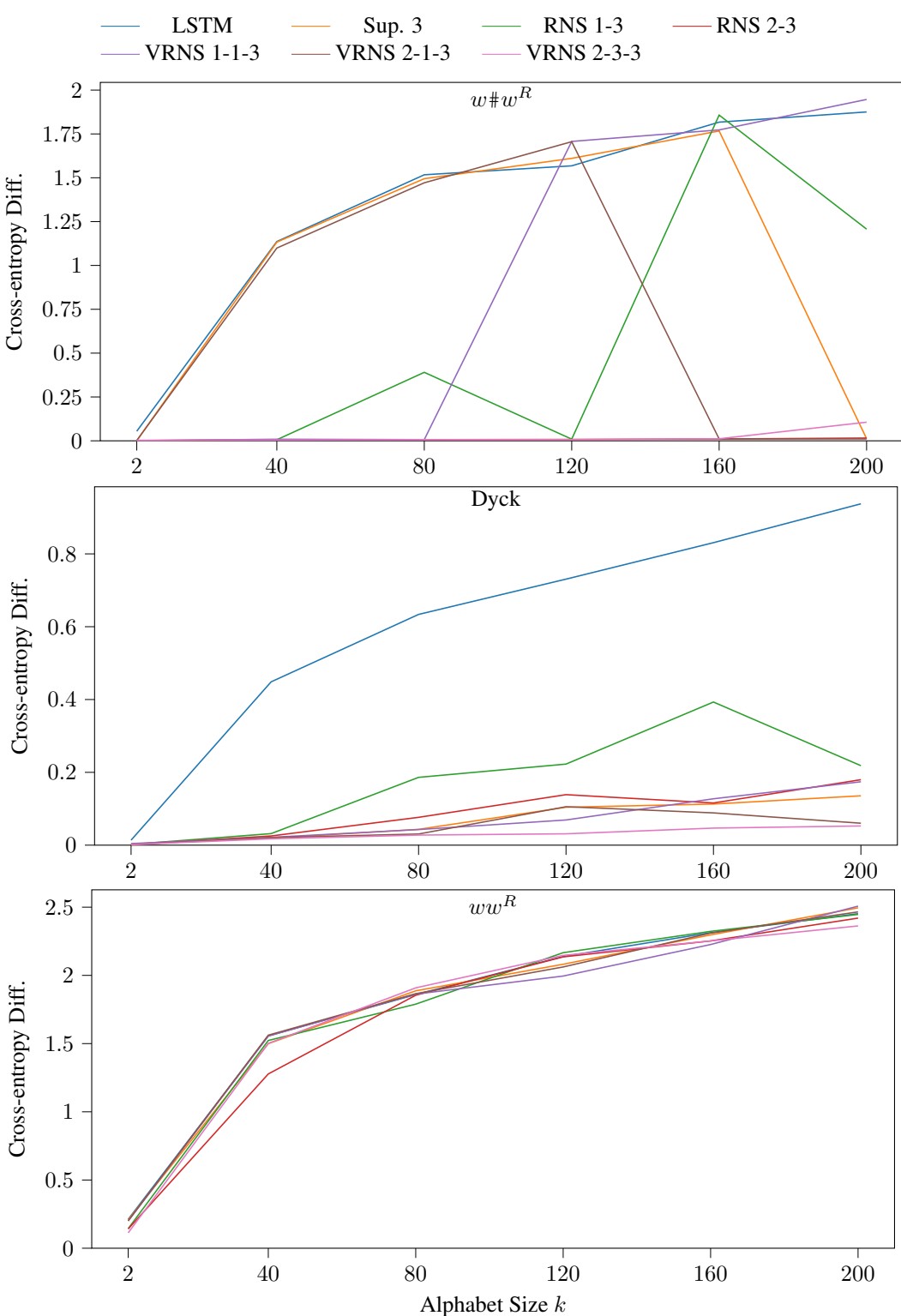

Figure 7: Best cross-entropy difference on the validation set vs. input alphabet size. On $w\#w^R$, surprisingly, only RNS 2-3 achieved optimal cross-entropy for all alphabet sizes. On the more complicated Dyck language, our new VRNS-RNN (VRNS 2-1-3, VRNS 2-3-3) achieved the best performance for large alphabet sizes. No models performed much better than the LSTM baseline on $ww^R$, although RNS 2-3 performed well for $k = 40$.

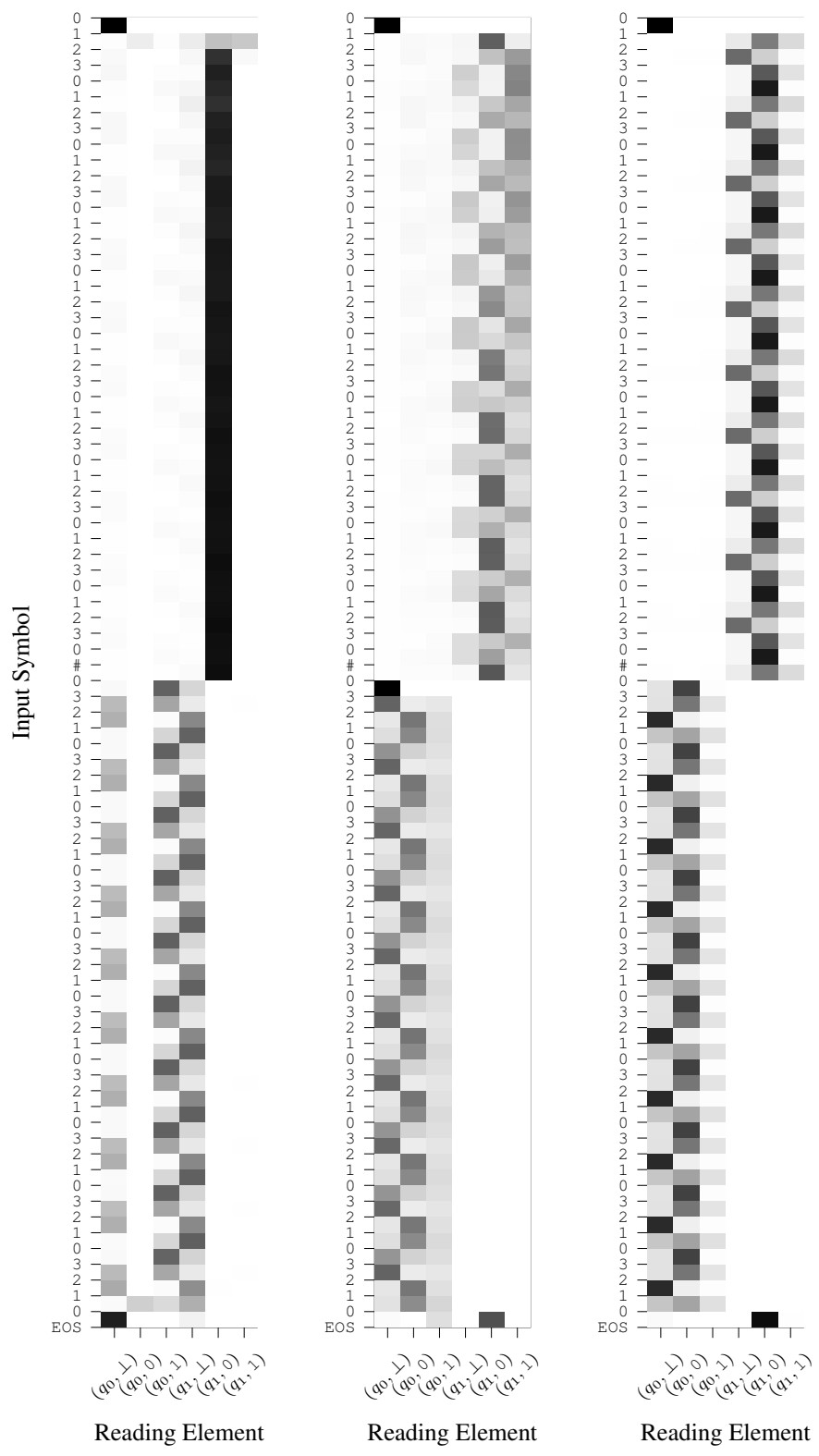

Figure 8: Heatmaps of $\mathbf{r}_t$ over time on a string from $w\#w^R$ when $k = 40$, generated from the best RNS 2-3 model (left) and two other random restarts (middle, right). The $w$ string repeats the pattern `0123`, which is clearly seen in the reading vectors. Black = 1, and white = 0.

