# OpenReview forum: "The Surprising Computational Power of Nondeterministic Stack RNNs"
_ICLR.cc/2023/Conference — ICLR 2023 poster_

### Official Review · Reviewer_HmX3 · 2022-10-24

**Confidence:** 2
**Correctness:** 4
**Technical Novelty And Significance:** 3
**Empirical Novelty And Significance:** 3
**Recommendation:** 8

**Clarity, Quality, Novelty And Reproducibility:**

- Very high quality.
- Mostly clear, though dense.
- This paper is easily reproducible if the authors release the code, but they do not state whether they plan to.
- I don't feel confident in my ability to assess the originality, but as far as I can tell it is a novel contribution to the space, as it analyzes very recently developed architectures.

**Strength And Weaknesses:**

Strengths:
- This paper is both clear and precise.
- The computational capacity of neural models in general has seen some increased interest lately.
- The result is interesting to me personally: that in theory as well as in practice, the nondeterministic stacks permit the learning of some non context free languages.
- The proposed architecture points to a possible future in which we continue to develop models inspired by discrete symbolic automata, but replacing more and more of them with continuous representations.


Weaknesses:
- This topic is very niche and might not attract much attention, because it’s such a specific analysis of the capacity of a very specific recent architecture. However, in general people have been paying more attention to questions of computational capacity in neural networks, as evidenced by how widely shared https://arxiv.org/abs/2207.02098 (Neural Networks and the Chomsky Hierarchy) recently was.
- I would like to see a little bit more discussion of the exact level of context sensitivity required for the languages being studied. Are they covered by particular indexed grammars? Can we characterize what particular classes of mildly context sensitive languages can be recognized?
- PTB, the only “realistic” dataset, is a very weak benchmark if you are using Mikolov’s approach, which has a high rate of unknown token masking and word-level tokenization.

Minor/Questions:
- It’s not really clear to me what you mean when you talk about needing multiple time steps for the deterministic case.
- It might be a good idea to briefly explain things like cross serial dependencies. I understand that the target audience probably knows these concepts, though.
- I felt like I would have benefited from a diagram during the extensive description of all of the different variables involved in these architectures.
- “Restricted form” doesn’t seem to be defined explicitly, and I’m not sure what makes these forms restricted.
- Admittedly, my automata theory is somewhat rusty. However, I recall that intersecting context free languages yields an undecidable problem (due to the post correspondence problem). Could you help me understand why undecidability isn’t an issue when you are proving that every intersection of a finite set of CFLs is recognizable by an RNS-RNN?


**Summary Of The Paper:**

This paper analyzes the capacity of stack-RNNs, with a particular focus on the existing architecture of the renormalizing nondeterministic stack RNN (RNS-RNN), which they contrast with deterministic stacks. First, they prove analytically that the RNS-RNN is capable of recognizing arbitrary context free languages. Then, they prove that the RNS-RNN is capable of recognizing *intersections* of sets of arbitrary CFLs, which is a language class that is not guaranteed to be context free.

They then look experimentally at the ability of stack RNNs to recognize 6 specific non-context free languages. They find that the nondeterministic stack architecture can recognize these languages with some additional explicit hints during training, and an architecture with multiple deterministic stacks can recognize these languages if given more hints. They develop a new architecture by using a stack of vectors instead of symbols, and show that it improves perplexity on penn tree bank and learns Dyck languages better.

**Summary Of The Review:**

This topic is very niche and might not attract much attention, because it’s such a specific analysis of the capacity of a very specific recent architecture. However, in general people have been paying more attention to questions of computational capacity in neural networks. The result is interesting to me personally: that in theory as well as in practice, the nondeterministic stacks permit the learning of some non context free languages.

---

> ### Author Response · Authors · 2022-11-11
> **On intersections of CFLs, extra timesteps, and paper clarifications**
>
> Dear Reviewer HmX3,
>
> Thank you very much for your constructive suggestions, questions, and interest in our work.
>
> **Response to Weakness 2:**  The hierarchy of intersections of context-free languages (CFLs) was studied by [Liu and Weiner (1973)](https://doi.org/10.1007/BF01762237). This hierarchy doesn't fit neatly into the hierarchy of indexed languages: for example, $\\{(ab^n)^n \mid n \geq 0\\}$ is not an indexed language ([Gilman, 1996](https://doi.org/10.1016%2F0304-3975%2896%2900244-7)), but it is the intersection of three CFLs, namely, $(ab^nab^n)^*(ab^*)^?$ and $ab^*(ab^nab^n)^* (ab^*)^?$ and $ab^n (ab^*)^{n-1} \cup \epsilon$.
>
> **Response to Weakness 3:** Yes, we use Mikolov's approach. See our response to Weakness 2 and Detail 5 of Reviewer AKsm.
>
> **Response to Minor Question 1:** Thank you for pointing this out -- we will try to clarify this in the main text of the paper. The reason why extra timesteps are expected to make a task (e.g. $w \texttt{\\#} w^R \texttt{\\#} w$ or $w \texttt{\\#}^n w$) easier to solve deterministically is described in Appendix C. Consider how a two-stack deterministic automaton would recognize the language $w \texttt{\\#}^n w$: While reading the first $w$, push it to stack 1. While reading the middle $\texttt{\\#}^n$, move the symbols from stack 1 to stack 2 in reverse. While reading the final $w$, match it with symbols popped from stack 2. The "extra" $n$ timesteps afforded while reading $\texttt{\\#}^n$ give a deterministic automaton exactly the number of steps needed to transfer all symbols from stack 1 to stack 2. On the other hand, in a task like $w \texttt{\\#} w$, an automaton must use a more complicated construction (see Appendix C under $w \texttt{\\#} w$) that involves three stacks, in which the automaton nondeterministically starts to transfer symbols between stacks halfway *before* reaching the $\texttt{\\#}$.
>
> **Response to Minor Question 2:** Thank you for this suggestion. We will try to address it as space allows. Is there something in particular that could be added to the bottom of page 4?
>
> **Response to Minor Question 3:** Thank you for this suggestion as well. Would a diagram illustrating the RNN controller-stack interface described in Section 2 be helpful in this regard?
>
> **Response to Minor Question 4:** The definition of restricted form is in the middle of the paragraph at the top of page 3. We will reword this section to make the definition of restricted form clearer and easier to spot.
>
> **Response to Minor Question 5:** You're right that intersections of CFLs flirt with undecidability, but there is a subtle distinction to be made. Deciding whether a given string belongs to the intersection of $k$ CFLs is decidable; it's deciding whether the intersection of $k$ CFLs is empty that is undecidable. The RNS-RNN only solves the former problem, not the latter, so undecidability is not an issue. See, for example, [Hoogeboom, "Undecidable Problems for Context-Free Grammars"](https://liacs.leidenuniv.nl/~hoogeboomhj/second/codingcomputations.pdf).
>
> **Comment on reproducibility:** Our code is already available on OpenReview via the supplementary material. We will release our code publicly on GitHub once the anonymity period is over (we briefly mention this at the end of the introduction).

---

> > ### Comment · Reviewer_HmX3 · 2022-11-11
> > **responses to questions**
> >
> > Thanks for your explanations. Are there more fine-grained classifications of the intersections you look at that can distinguish learnable from unlearnable intersections? Are all intersections of context free languages learnable, by your proof?
> >
> > Yes, I think a diagram illustrating the controller stack interface could be helpful and provide context for the variable names used throughout the paper.

---

> > > ### Author Response · Authors · 2022-11-11
> > > **On learnability of intersections of CFLs**
> > >
> > > Our proof shows that all intersections of CFLs are *expressible* by an RNS-RNN in the sense that a solution with perfect recognition accuracy always exists, assuming the RNS-RNN is sufficiently large and weights of $-\infty$ are allowed in certain parts of the network. Whether all intersections of CFLs are *learnable* in the sense that the solution is practically reachable through gradient descent is a question we do not explore theoretically. But empirically we did see, for example, that the RNS-RNN was able to learn $w\texttt{\\#}w^R\texttt{\\#}w$ but not $ww^Rw$, though both are intersections of two CFLs.

---

### Official Review · Reviewer_ACch · 2022-10-25

**Confidence:** 3
**Correctness:** 4
**Technical Novelty And Significance:** 3
**Empirical Novelty And Significance:** 3
**Recommendation:** 6

**Clarity, Quality, Novelty And Reproducibility:**

The authors made strong efforts to ensure the reproducibility of the results reported in the paper.

The paper is a follow-up of DuSell and Chiang, and is thus not entirely original. But the theoretical results are new. The vector stack is also new.

The paper is readable, but the presentation of nondeterministic stack RNN can be improved with the help of diagram illustration and simplified instances. Some background on formal language can be helpful.

**Strength And Weaknesses:**

Strengths:

(1) The paper is theoretically solid. The analysis of the ability of nondeterministic stack RNN in recognizing formal language is a useful theoretical contribution.

(2) The proposed stack of vectors is a useful improvement of nondeterministic stack RNN.

Weaknesses:

(1) The paper relies heavily on the previous paper of DuSell and Chiang, and is not entirely novel.

(2) The experiments mostly compare with baseline LSTM, which may not be a strong baseline.

(3) No discussion of related models such as neural Turing machine.

(4) It is unclear in what sense the model understands natural language. Can it do better than transformer model in terms of language understanding?

**Summary Of The Paper:**

This paper analyzes the properties of nondeterministic stack RNN, and proposes a new version with stacks of vectors. (1) The paper shows that nondeterministic stack RNN can recognize non-context-free languages. (2) The stack RNN with small stack alphabet can recognize languages with a much larger alphabet. (3) RNN with stacks of vectors show improved performance.

**Summary Of The Review:**

The paper makes a solid theoretical contribution. The new version with stack of vectors is empirically useful.

---

> ### Author Response · Authors · 2022-11-11
> **On novelty, baselines, neural TMs, NLU, and clarifications**
>
> Dear Reviewer ACch,
>
> Thank you very much for your thoughtful review, comments, and helpful suggestions.
>
> **Response to Weakness 1:** Although this paper does build on DuSell and Chiang (2020, 2022), we believe that the two new theoretical results, the empirical results on non-CFLs, the empirical results on CFLs with large alphabets, the introduction of our novel VRNS-RNN architecture, and the results on natural language all constitute a substantial original contribution.
>
> **Response to Weakness 2:** In our experiments, we compare the RNS-RNN and VRNS-RNN against multiple variants of the competitive superposition stack RNN baseline, including a multi-stack version that, as far as we know, is new to this paper. The inclusion of the LSTM is meant to provide a reference point for the performance of the RNN controller without an external stack.
>
> **Response to Weakness 3:** Thank you for this suggestion; we will try to add this discussion to the paper as space allows. The neural Turing machine uses a tape with a fixed number of memory cells that does not grow dynamically with the input sequence, so theoretically it is no more powerful than a finite automaton, in contrast with the RNS-RNN, which, as this paper shows, is more powerful than a pushdown automaton.
>
> **Response to Weakness 4:** The motivation for studying the RNS-RNN and developing the VRNS-RNN is based on the connection between context-free languages and the compositional nature of natural language syntax, and so this paper focuses on formal language theory and syntax, rather than natural language understanding. We agree that the question of whether the improved syntactic processing of the (V)RNS-RNN aids understanding of natural language compared to the transformer is an important one, though outside the scope of this paper.
>
> > the presentation of nondeterministic stack RNN can be improved with the help of diagram illustration and simplified instances
>
> Thank you for this suggestion. Would a diagram illustrating the RNN controller/stack architecture described in Section 2 be helpful in this regard?
>
> > Some background on formal language can be helpful.
>
> Thank you for this suggestion as well, and we will try to address it as space allows. Are there particular points you believe would benefit from more explanation?

---

### Official Review · Reviewer_AKsm · 2022-10-26

**Confidence:** 3
**Correctness:** 3
**Technical Novelty And Significance:** 3
**Empirical Novelty And Significance:** 2
**Recommendation:** 8

**Clarity, Quality, Novelty And Reproducibility:**

- The paper is clearly written but can be more self-contained.
- The experiments are well executed.
- Enough experimental details are provided to reproduce the results.

**Strength And Weaknesses:**

Strength:
- The paper is fairly clear for how dense it is.
- The experiments are well-designed to support the claims
- It provides new insights into the modeling capacity of stack RNNs.

Weaknesses:
- The paper can be more self-contained. It heavily builds on the RNS-RNN work; for readers unfamiliar with this previous work (like myself), several details need to be verified to examine some of the claims.
- The results are too weak to back up the claim that “nondeterministic stack RNN can recognize many non-context-free languages.”
- The empirical contribution is not the paper’s strength. The results are on synthetic experiments or tiny-scale natural language tasks. This could limit the paper’s impact.

Details:
- I appreciate the detailed description of RNS-RNN; but there are several details I want to double-check with the authors. Regarding the transition weight function above Eq. 1: can the authors confirm that the transition weight function conditions on the state of the stack, through $\mathbf{h}_t$?
- I assume each row of $\mathbf{W}_a$ can be seen as a vector representation of an action. For example, are “push(x)” and “push(y)” considered different actions and get different representations in $\mathbf{W}_a$?
- Following the above: if “push(x)” and “push(y)” get different vector representations, each symbol is already embedded into a vector representation. If this is the case, can the authors comment on why it is surprising that RNS-RNN can recognize CFLs with large alphabets?
- How does the theoretical result in Section 3 connect to existing conclusions on real-time PDA and context-free languages ([1] and the older works cited therein)?
- Regarding the results in Figure 1: even the best stack RNN variant (RNS 3-3) achieves mixed results compared to the LSTM baseline (outperforms in 2 languages and underperforms in others). The framing is too strong that stack RNNs can recognize “many” non-context-free languages.
- I would appreciate it if the paper could include an evaluation of non-context-free languages with large alphabets.
- Missing references: using vector states in automata was explored in [2] and [3]

[1] https://arxiv.org/pdf/1302.1046.pdf

[2] https://arxiv.org/abs/1805.06061

[3] https://arxiv.org/abs/1808.09357


**Summary Of The Paper:**

This paper examines the capacity of renormalizing nondeterministic stack RNN (RNS-RNN), a stack RNN variant proposed by previous work, and proves that it can recognize all context-free languages. It empirically shows that, surprisingly, it performs well in recognizing non-context-free languages, even when the language’s alphabet is larger than the stack’s. To increase its capacity, the paper proposes a new variant that embeds the stack states into vectors and shows that it achieves competitive performance in recognizing large-alphabet formal languages.


**Summary Of The Review:**

This paper provides new insights into stack RNNs. I vote for acceptance.

---

> ### Author Response · Authors · 2022-11-11
> **Response to Reviewer AKsm, part 1 of 2**
>
> Dear Reviewer AKsm,
>
> Thank you very much for your thoughtful review, suggestions, and detailed follow-up questions.
>
> **Response to Weakness 1:** Thank you very much for the suggestion. We will try to address this as space allows. Two other reviewers recommended adding a conceptual diagram of the model -- do you believe it would be helpful to add a diagram of the RNN controller-stack interface described in Section 2? Are there any details of the RNS-RNN in particular that would benefit from a more detailed explanation?
>
> It may be helpful to point out that only Equation 1 is required in order to understand the RNS-RNN mathematically. Equations 2-7 merely describe the algorithm for computing Equation 1 tractably. We include Equations 2-7 for two reasons: (1) to give the full form of the equations following our minor modifications described in Appendix A, and (2) to provide context that is required when describing the implementation of the VRNS-RNN in Equations 9-14.
>
> **Response to Weakness 2 and Detail 5:** We believe the empirical results show, across multiple tasks, that multi-stack RNNs and RNS-RNNs are capable of recognizing a non-context-free phenomenon (namely, cross-serial dependencies) that is of interest to both linguistics and formal language theory. We would be happy to rephrase part of our paper to clarify this.
>
> Regarding the performance of stack RNNs relative to the LSTM baseline in Figure 1, on $w\texttt{\\#}w^R\texttt{\\#}w$, the results for stack RNNs are actually very positive except with regard to length generalization, which is poor. The RNS-RNN achieved optimal performance on the validation set and on the test set for strings within the length range it was trained on ($[40, 80]$), but it generalized poorly on lengths greater than 80. Sup. h also seems to have had a similar issue, although it did not even attain optimal validation performance. The RNS-RNN also seems to have had length generalization issues on $\texttt{a}^n \texttt{b}^n \texttt{c}^n$ and to a lesser degree on $ww$ and $w \texttt{\\#}^n w$. We hope to explore regularization techniques to alleviate this in future work. As for $ww$, we consider this to be a negative result for all models, given that their validation cross-entropy is so far from optimal. We believe the more important message is that $ww$ (and $ww^Rw$ in Appendix F) is an example of a language that none of the models can recognize. We will update our paper to clarify all of these points in the discussion of the results.
>
> **Response to Weakness 3:** We agree that the formal language tasks are largely of theoretical interest, but we believe this work represents an important step from theory towards more practical applications, and the results on the Penn Treebank, if a relatively small task, are encouraging in this regard.
>
> **Response to Detail 1:** Yes, the action weights $a_t$ a.k.a. $\Delta[t]$ depend on the hidden state $\mathbf{h}_t$, which depends on the stack reading $\mathbf{r}_t$, which is a summary of all nondeterministic PDA runs in the form of a marginal distribution over their current states and top stack symbols.
>
> **Response to Details 2 and 3:** Yes, this is a useful insight that applies, we think, not so much to $\mathbf{W}\_{\mathrm{a}}$ directly, but to the tensor of transition weights $\Delta[t]$, which is computed by $\mathbf{W}_{\mathrm{a}}$. To expand on your $\mathrm{push}(y)$ example, it is accurate to say that for each stack symbol type $y$, there is a slice of $\Delta[t]$ that consists of push weights for all triples $(q, x, r)$, where $q$ and $x$ are the state and top symbol of runs where a particular push transition will take effect, and $r$ will be the new state after applying the transition. However, it is not clear how this vector could be encoded in the distribution over runs in a way that could be recovered by popping later. The result that RNS-RNNs can simulate large alphabets is surprising in that, whatever strategy is employed, this use of the stack WFA is neither intended nor particularly intuitive, and impossible for traditional weighted PDAs (upon which the RNS-RNN is based), because PDAs cannot coordinate runs. Even though RNS-RNNs have this ability, the development of the VRNS-RNN is important because increasing the stack embedding size $m$ is much more scalable computationally than increasing $|Q|$ or $|\Gamma|$.
>
> **Response to Detail 4:** We're not sure what the exact relevance of the cited paper by Silva is; if the relevant point is that real-time PDAs recognize all CFLs, [Greibach (1965)](https://dl.acm.org/doi/pdf/10.1145/321250.321254) showed that real-time nondeterministic PDAs recognize all CFLs. The RNS-RNN's *restricted form* of PDA is a more restricted subset of real-time PDAs. To our knowledge we are the first to show they too recognize all CFLs. If there is another point you would like us to comment on, could you please clarify?
>
> (continued in next post)

---

> > ### Author Response · Authors · 2022-11-11
> > **Response to Reviewer AKsm, part 2 of 2**
> >
> > (continued)
> >
> > **Response to Detail 6:** Thank you for this suggestion. We agree that it would be interesting to see if the RNS-RNN can employ both of its "tricks" at the same time. If we have time, we will try to include some experiments on non-CFLs with large alphabets in a later revision and get back to you.
> >
> > **Response to Detail 7:** Thank you; we will cite these papers.

---

> > > ### Comment · Reviewer_AKsm · 2022-11-18
> > > **After author response**
> > >
> > > Thanks for the response! The paper is definitely in better shape now. Hence I will increase my score to 8.
> > >
> > > Re detail 4: your response on this one is exactly what I was looking for, which is already included in the paper.

---

### Official Review · Reviewer_Gwva · 2022-10-28

**Confidence:** 3
**Clarity, Quality, Novelty And Reproducibility:** The clarity, quality, novelty and rep…
**Correctness:** 4
**Technical Novelty And Significance:** 3
**Empirical Novelty And Significance:** 3
**Recommendation:** 6

**Details Of Ethics Concerns:**

None.

**Strength And Weaknesses:**

Stengths:

-  The paper is clearly presented, and investigates an interesting phenomena.  Particularly, the comparison of neuro-symbolic models' representational strength using formal languages is a promising approach, which is currently under-explored.
-  The theoretical analysis is thorough,
-  The empirical analysis is convincing, and provides good motivation for why the vector-valued stack should increase the model's power (essentially by allowing the model to choose the number of symbols stored on the stack).

Weaknesses:

-  The theoretical analysis is essentially the translation of a pushdown automata (PDA) construction into a RNS-RNN, which in itself is not surprising (although useful for determining the dimensions of an RNS-RNN equivalent to a given PDA strength).
-  It is unclear if there is a specific advantage to using vector RNS-RNNs with its stack memory model in terms of performance as opposed to say models with access to a key-value based memory model, as in a DQN model.  It would be interesting to see such comparisons, both on the formal and natural language data (for the former, one would expect that there are limitations on representational strength imposed by the number of possible keys, but it is not clear how non-determinism would affect this).

**Summary Of The Paper:**

The paper presents a part empirical / part theoretical analysis of the representational power of non-deterministic stack RNNs.  It is shown theoretically that renormalizing nondeterministic stack RNNs  (RNS-RNNs) have the power to recognize context-free languages and their intersections (Props 1 and 2).  The ability of RNS-RNNs to recognize context-free languages is investigated empirically, whose performance is shown to surprisingly exceed what might be expected theoretically.  The authors show that the model is using the non-deterministic state of the stack to exceed the limitations that would otherwise exist for a deterministic model (Fig. 3).  This prompts the introduction of a novel architecture, where the stack is allowed to store real vectors rather than only discrete symbols, and it is shown that this representation achieves SOTA performance when modeling real language data in terms of perplexity (Table 1).

**Summary Of The Review:**

An interesting investigation of the representational strength of a neuro-symbolic model, combining theoretical and empirical analysis.  The work will be of interest to those working on formal languages, deep-learning and natural language processing.

---

> ### Author Response · Authors · 2022-11-10
> **On restricted PDAs and DQNs**
>
> Dear Reviewer Gwva,
>
> Thank you very much for your constructive comments and thoughtful review.
>
> **Response to Weakness 1:**  It is true that the conversion of a PDA in restricted form to an RNS-RNN is relatively straightforward (although the full proof did require working out some important details in the RNN controller). The main contribution of the proof of Proposition 1, rather, is that any CFL can be recognized by a PDA in restricted form (as opposed to the usual textbook definition of PDA), which had not previously been shown.
>
> **Response to Weakness 2:** Our familiarity with DQNs is limited, but we see two main advantages of the VRNS-RNN compared to a reinforcement learning approach: (1) simpler training (2) theoretical coverage of all CFLs (including nondeterministic ones).
>
> With regard to advantage (1), because the VRNS-RNN's stack module is differentiable, the VRNS-RNN can be trained end-to-end with standard backpropagation and gradient descent techniques, which allows it to be easily connected to other differentiable modules (e.g. as part of a neural machine translation system) and trained jointly with them. Moreover, as discussed by DuSell and Chiang (2022) regarding the RNS-RNN, because the stack reading is the sum of all possible PDA runs, the best run (or runs) always contributes some signal to the loss function, making it easier to find with gradient descent.
>
> As for advantage (2), although it may be possible to train a DQN to operate a deterministic stack, such a model would not be capable of recognizing all CFLs, as many require nondeterminism (such as $ww^R$, $\\{ \texttt{a}^i \texttt{b}^j \texttt{c}^k \mid i = j \text{ or } j = k \\}$, and the hardest CFL ([Greibach, 1973](https://dl.acm.org/doi/10.1137/0202025))). If one were to train a DQN to operate a *nondeterministic* stack, in the worst case, recognizing a CFL might require an exponential number of parallel runs, in which case a dynamic programming algorithm such as the one employed in the VRNS-RNN would be required anyway. On the other hand, DQNs might be an attractive alternative in scenarios that require no or limited nondeterminism.
>
> As for the use of a key-value store, note that many patterns, such as $w\texttt{\\#}w^R$ and $w \texttt{\\#} w$, require *positional* information to recognize them, so a content-based key-value store may be of limited use in tracking the syntactic structure of a sequence compared to a stack (imagine using a Python dict to reverse a string).
>
> **Comment:** The language modeling results on the Penn Treebank are not SOTA, but they do show that our VRNS-RNN attains better perplexity than the LSTM, superposition stack RNN, and RNS-RNN. These experiments are meant to show the relative performance of the models on a natural language task (albeit a relatively simple one), similar to [Joulin and Mikolov (2015)](https://arxiv.org/abs/1503.01007), [Yogatama et al. (2018)](https://openreview.net/forum?id=SkFqf0lAZ), and [DuSell and Chiang, 2022](https://openreview.net/forum?id=5LXw_QplBiF).

---

> > ### Comment · Reviewer_Gwva · 2022-12-13
> > **Response**
> >
> > Thanks to the authors for their detailed response.  Thanks particularly for clarifying the contributions/contents of Prop. 1.  I'm unsure if the DQN is in principle less expressive than models with a non-deterministic stack-based memory though - it seems that it should be possible to learn to mimic a stack using a key-value store by treating it as a linked list, and since the DQN can read back a weighted combination of key-value pairs this would seem to mimic a non-deterministic stack, and positional information can be implicit in the structure of the keys.  These might be interesting issues to clarify further, along with the potential advantages regarding training efficiency mentioned in the response.

---

### Author Response · Authors · 2022-11-18
**Summary of paper revisions**

Dear reviewers,

We have made the following revisions to our paper:

* We have added a new figure (Fig 1) that illustrates the RNN controller-stack interface described in Section 2 ("Stack RNNs").
* We have reworded parts of Section 2.2 ("Renormalizing nondeterministic stack RNN") for clarity.
  * We have slightly reworded the definition of restricted form to make it more explicit.
  * We now point out to the reader that only Equation 1 is necessary for describing the RNN mathematically, and that the rest of the section may be skipped unless one is interested in implementation details of the dynamic programming algorithm. This is meant to reduce the burden of understanding the technical details in this section before moving on to the rest of the paper.
* In section 3 ("Recognition power"), we have added more discussion relating our theoretical results to prior work.
  * We now point out that although Siegelmann and Sontag (1992) showed that RNNs are as powerful as Turing machines, this result assumes infinite precision and unlimited extra timesteps, which do not hold true in practice, and that our proofs do not rely on unlimited extra timesteps.
  * We point out that neural Turing machines have a tape of fixed size that does not grow with the length of the input string, so they are theoretically no more powerful than finite automata.
  * We reference previous work by Stogin et al. (2020) showing that any real-time DPDA can be stably encoded in a variant of the superposition stack.
  * We relate our Proposition 1 to the long-known result that real-time PDAs are no less powerful than general PDAs (Greibach, 1965), emphasizing our contribution of showing that not only real-time PDAs, but also real-time PDAs in restricted form, are no less powerful than general PDAs.
* We have changed Definition 1, which defines recognition for RNNs, in order to simplify details in the proofs in Appendix B. Acceptance is now determined by the output of an MLP.
* We have removed a sentence at the end of Section 3 that incorrectly stated that DCFLs are closed under intersection.
* In Section 4 ("Non-context-free languages"), we clarify why the presence of extra timesteps in $w \texttt{\\#} w^R \texttt{\\#} w$ is expected to make learning easier, and we refer the reader to the Appendix for more details.
* In Section 4, we point out that RNS 3-3 does not generalize well to unseen lengths for $w \texttt{\\#} w^R \texttt{\\#} w$.
* In Section 5 ("Capacity"), we mention related work by Schwartz et al. (2018) and Peng et al. (2018).
* We have added a few missing details, corrections, and more detailed explanations to Appendix B.1 ("Proof of Proposition 1").
  * We have added a bottom symbol $\bot$ to our definition of PDA, which makes the connection to the RNS-RNN more direct. We now require the top of the stack to be $\bot$ for the PDA to accept its input.
  * In the proof for Lemma 4, we have added a detail about adding an initial non-scanning transition from $q_0$ to $q_{\mathrm{loop}}$, which had been mistakenly omitted.
  * In the same section, we have removed a superfluous case handling rules of the form $A \rightarrow a A_1$.
  * In the same section, we fixed a typo in the construction for rules of the form $A \rightarrow a b B_1 \cdots B_p$, so the second transition scans $b$ instead of $a$.
  * In the same section, we have added more intuitive explanations for each of the constructions used in the conversion from 2-GNF rules to PDA transitions.
  * In the proof for Lemma 6, we have clarified that the transition weights depend on the input symbol $w_t$.
  * The construction in Lemma 6 no longer relies on computing $\frac{1}{b_n}$, making it simpler.
* We have added a correction and some more detailed explanations to Appendix B.2 ("Proof of Proposition 2").
  * We have modified the proof to accommodate the inclusion of $\bot$ in the definition of PDA.
  * We have fixed a mistake that allowed the RNS-RNN to accept $w$ if $w \in L_1$ but $w \not\in L_2$, or vice versa.

We have also uploaded a new version of the code in the Supplementary Material that fixes a NameError.

---

### Decision · Program_Chairs · 2023-01-20

**Decision:**

Accept: poster

**Justification For Why Not Higher Score:**

N/A

**Justification For Why Not Lower Score:**

N/A

**Metareview: Summary, Strengths And Weaknesses:**

The paper presents a surprising ablility of the nondeterministic stack RNNs, which is it can learn not only CFLs but non-CFLs. It can
recognize languages with much larger alphabet sizes than one might expect given the size of its stack alphabet. Finally, to increase the information capacity in the stack and allow it to solve more complicated tasks with large alphabet sizes mbols. The model achieves a siginificant improvenment in experiments conducted in PTB. The reviewers agree that this is a solid contribution to the community. I would recommend acceptance of this paper to ICLR.

**Note From Pc:**

if the above contains the word "oral" or "spotlight" please see: "oral" presentation means -> notable-top-5% and "spotlight" means -> notable-top-25%. As stated in our emails, we are disassociating presentation type from AC recommendations